# Real-space machine learning of correlation density functionals

Elias Polak ⓘ, Heng Zhao ⓘ & Stefan Vuckovic ⓘ ✉

Machine learning (ML) plays a pivotal role in extending the reach of quantum chemistry methods for simulating both molecules and materials. However, leveraging ML to overcome the limitations of human-designed density functional approximations (DFAs), the primary workhorse for quantum simulations, remains a major challenge due to their severely limited transferability to unseen chemical systems. Here, we demonstrate how transferability is achieved using real-space ML, where energies are learned point by point in space through energy densities. Central to our real-space learning strategy is the derivation and implementation of correlation energy densities from regularized perturbation theory. This enables two key advances toward constructing highly transferable DFAs, grounded in the Møller-Plesset adiabatic connection framework, for correlation energies defined with respect to the Hartree-Fock reference. First, we introduce the Local Energy Loss, whose data efficiency (expanding each system's single energy into thousands of data points) dramatically enhances transferability when combined with a physically informed ML model. Second, we formulate a real-space, machine-learned, and regularized extension of Spin-Component-Scaled second-order Møller-Plesset perturbation theory, yielding transferable DFAs that effectively mitigate the self-interaction errors common to traditional DFAs.

Machine learning (ML) is driving a paradigm shift across scientific disciplines, including quantum chemistry (QC), where it reshapes the landscape of used methods[1–6]. The recent surge in ML has further increased the importance of density functional approximations (DFAs), already a cornerstone of quantum simulations in materials science and chemistry (see the recent review by von Lilienfeld and co-workers[7]). On one hand, DFAs generate vast amounts of data to train ML models[7–15], greatly extending their reach in time and length scales[16–25]. On the other hand, ML techniques provide improved DFAs[26–38] (DFAs), with the DM21 functional by DeepMind[39] being a prominent example.

A remaining critical problem with ML in QC is their limited transferability: the ability to generalize to unseen data[6,40–42]. These limitations hinder the applicability of ML models in QC, making users cautious and often leading them to stick with well-established *old-school* methods[43] over *new-school* ML counterparts. This situation has

lead to a *no man's land* between old-school and new-school DFAs[44]: the promised revolution of ML-based DFAs is hampered by the far broader applicability of old-school DFAs[44], such as B3LYP[45–48] or PBE[49]. For example, while DM21's training on systems with fractional charges and spin addresses some limitations of old-school functionals[39], its catastrophic transferability failures have recently become evident[44]. Namely, it has been shown that DM21 does not converge for certain transition metal (TM) atoms, a convergence task easily handled by reputable old-school functionals[44]. Thus, it is no wonder that organic chemists still prefer old-school DFAs over DM21 or other new-school models for TM-catalyzed reaction mechanisms[50], despite major TM shortcomings of the former[51,52].

To move from this no man's land and leverage the power of ML for DFAs design, we need to solve the underlying transferability problem. Helping strategies are the use of physical constraints[53–56] (a mix of old- and new-school methods), more diverse data in the training set[40], or

Department of Chemistry, University of Fribourg, Fribourg CH-1700, Switzerland. ✉e-mail: stefan.vuckovic@unifr.ch

the engineering of new features[57]. Yet, an angle in ML of DFAs that requires more attention is the training data efficiency, particularly since feeding more data to ML models can become a never-ending game due to the data hunger in ML models and the vastness of chemical space[2,3,5]. The purpose of this work is to critically examine data efficiency in training DFAs and maximize it in order to embed transferability in ML of DFAs.

The primary goal when training DFAs is to learn how a given electronic density translates into energy. However, ML practices in QC currently treat energies as not very informative, following a 1 system = 1 energy data point approach (see refs. 30–32,34,37,39,54,58,59). For example, when learning force fields, forces (energy gradients) are much more informative than energies[19,60]. Similarly, recent works have shown that in current ML DFA practices, electronic densities are also far more informative than energies[31,34,54,58,59] (each grid point is a density datapoint). While using electronic densities effectively enhances the transferability of ML-based DFAs, applying a point by point learning strategy to energies (typically the primary target of simulations) offers a distinct direction for improving ML-based DFAs. Identifying transferability as the key issue in ML DFAs, here we establish a framework for making energy training more data-efficient and address the challenges that must be overcome to leverage this data efficiency to embed transferability into ML DFAs.

In view of our objective to enhance data efficiency for energies in ML-based DFAs, we introduce real-space energy learning and apply it to machine-learn a DFA for correlation energy (a crucial target for DFAs). This approach expands each system's single energy data point into thousands of energy data points for the training. During the learning of our DFAs, each point in space contributes to the loss function, which we call Local Energy Loss (LES). Crucially, LES penalizes error cancellations between energy contributions from different regions in space, thereby enhancing transferability. With LES, every system in the training set becomes an entire energy dataset, and we show here that a careful, physically-informed application of LES is essential to fully realize its data-efficiency potential and to embed transferability in ML of DFAs.

Applying LES for ML of correlation DFAs requires two crucial ingredients: (i) a well-defined correlation energy contributions at each point in space $e_c(\mathbf{r})$ (i.e., correlation energy density per particle, see Eq. (1) below) and (ii) a robust strategy for generating accurate $e_c(\mathbf{r})$ training data. To meet these requirements simultaneously, we develop here LES-based ML DFAs for correlation energies defined with respect to the Hartree-Fock[61,62] (HF) reference. While historically DFA development has been tied to Kohn-Sham density functional theory[63] (KS DFT), recent theoretical advances based on the Møller-Plesset adiabatic connection (MPAC) formally ground the developments of correlation DFAs evaluated on HF densities (see ref. 64). Constructing DFAs on fixed (HF) densities within the MPAC framework enables us to isolate and focus specifically on real-space energy learning strategies, complementing (see Discussion) existing real-space density learning approaches[31,34,54,58,59]. As this work employs local energy quantities, we note that these quantities provide valuable chemical insights when well-defined (see, e.g., refs. 65–67). Since $e_c(\mathbf{r})$ is not uniquely defined, here we adopt a physically transparent definition arising from the MPAC framework[64] and demonstrate its advantages for LES.

An overview of the key methods presented in this paper is given in Fig. 1. To demonstrate the power of the LES-based approach, and more generally real-space ML for DFAs, we first construct a robust proxy reference for $e_c(\mathbf{r})$ that preserves the original MPAC-based definition and efficiently implement it to enable direct training of our ML models for $e_c(\mathbf{r})$. This proxy reference $e_c(\mathbf{r})$ is built by combining second-order perturbation theory[64] (PT2) [magenta circle in Fig. 1(a)] and the specific PT2 regularization[68,69] [blue square in Fig. 1(a)], which is crucial for making our proxy reference sufficiently accurate. The effect of regularization on the PT2's $e_c(\mathbf{r})$ for the helium dimer (going from the

magenta circle to the blue square in Fig. 1(a) is displayed in Fig. 1(d, top). We implement a numerical data generator of this proxy reference $e_c(\mathbf{r})$ by leveraging modern Python libraries (e.g., JAX[70]). Input features tailored to the problem, such as Grimme's real-space electronic correlation measures[71,72], enable us to construct a robust neural network (NN) for $e_c(\mathbf{r})$ [Fig. 1(b)]. We then contrast the transferability of our LES strategy with the common global energy loss (GES) that adopts standard 1 system = 1 energy data point approach [cyan and maroon diamonds in Fig. 1(a)]. Keeping other factors in the DFA training the same allows us to isolate how the transferability is affected when we move from GES to LES [Fig. 1(c) shows how well the two models trained on small atoms transfer to the dissociation curve of BH within spin-restricted calculations]. Similar transferability tests reveal subtle yet crucial requirements for successful and robust LES applications: it should be defined in terms of $e_c(\mathbf{r})$ rather than alternative quantities (e.g., its density-weighted counterpart), and coupled with a physically-informed ML model trained on a physically-motivated $e_c(\mathbf{r})$ definition. We also derive the contributions at each point in space for different spin channel pairs of our proxy $e_c(\mathbf{r})$ [purple and orange circles or squares in Fig. 1(a)] and we show these spin-resolved interaction components for the helium dimer in Fig. 1(d, bottom). We then use spin-resolved energy densities per particle to build a real-space, machine-learned and regularized extension of spin-component-scaled[73,74] (SCS) PT2 correlation energy (the performance of this ML strategy for the formic acid dimer is shown in Fig. 1(e, top)). The resulting model [green diamond in Fig. 1(a)] allows the construction of DFAs bridging the gap between our proxy reference correlation energies (regularized PT2) and their exact counterpart.

## Results

### Local and global energy loss (LES vs GES)
Distinguishing between LES and GES is a crucial point of this work when training ML DFAs. To define LES and GES generally, consider the reference (i.e. exact) energy

$$E^{\mathrm{ref}} = \int e^{\mathrm{ref}}(\mathbf{r})\rho(\mathbf{r})d\mathbf{r} \tag{1}$$

with a corresponding reference energy density[57] per particle, $e^{\mathrm{ref}}(\mathbf{r})$, and electronic density $\rho(\mathbf{r})$. An ML energy quantity defined in the same way is indicated using the ML superscript. Then, GES reads

$$\mathcal{L}_{\mathrm{GES}} \sim \left| E^{\mathrm{ref}} - E^{\mathrm{ML}} \right|. \tag{2}$$

In contrast, with LES, we consider the pointwise difference of the reference and ML energy densities per particle weighted by the density:

$$\mathcal{L}_{\mathrm{LES}} \sim \int \left| e^{\mathrm{ref}}(\mathbf{r}) - e^{\mathrm{ML}}(\mathbf{r}) \right| \rho(\mathbf{r})\,d\mathbf{r}. \tag{3}$$

Minimizing LES, strictly defined in terms of the energy density per particle, instead of GES turns each point in space into an energy data point, and as we shall see, moving from GES to LES dramatically enhances the transferability of the underlying ML DFA even with very small training sets.

### Improving and deriving PT2 correlation energy densities per particle
To demonstrate the difference between LES and GES, we will target correlation energy approximations. While correlation DFAs are typically developed within the KS DFT framework, recent work has shown that the Møller−Plesset adiabatic connection (MPAC) formally grounds the construction of DFAs mapping Hartree−Fock (HF) densities directly to correlation energies (defined here w.r.t. the HF energies). As

 

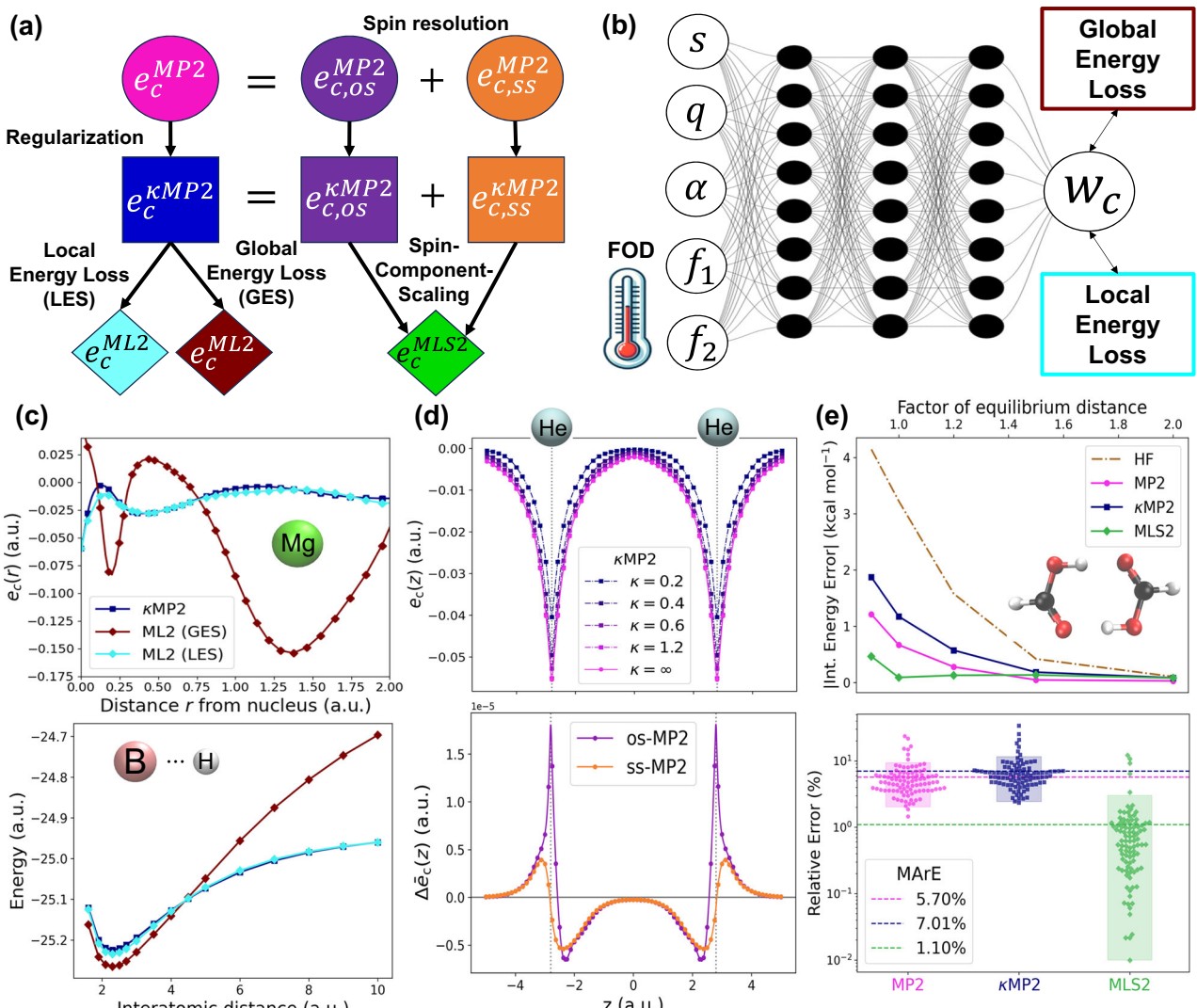

**Fig. 1 | Real-space machine learning correlation energy densities. a** Overview of methods and their connections. The correlation energy density per particle $e_c$ of second-order Møller-Plesset perturbation theory[83] (MP2) is spin-resolved into opposite-spin (os) and same-spin (ss) parts, and $\kappa$-regularized ($\kappa$MP2). Real-space machine-learning strategies presented here are energy densities from $\kappa$MP2 (ML2) and extensions of spin-component-scaled $\kappa$MP2 (MLS2). **b** Neural network illustration[123] with real-space features defined in Supplementary Section S6 in the SI (reduced density gradient $s$, reduced density Laplacian $q$, regularized energy kinetic variable $\alpha$ and temperature-dependent fractional occupation number weighted densities[71,72] (FOD) $f_1$ and $f_2$), and LES/GES strategies to produce machine-learned weights $w_c$ from Eq. (14). (**c, top**) Correlation energy density per particle of the Mg atom for GES-based and LES-based ML2, and $\kappa$MP2 as a proxy reference.

(**c, bottom**) Corresponding dissociation curve of the BH diatomic system. (**d, top**) Correlation energy density per particle of the helium dimer at an interatomic distance of 5.6 a.u., plotted along the principal axis of the system for different values of the $\kappa$-regularization. (d, bottom) Spin-resolved interaction correlation energy density of the helium dimer (dimer energy density minus that of atoms). (**e, top**) Errors in interaction energies (kcal mol$^{-1}$) along the dissociation of the formic acid dimer (geometries and reference values are taken from the S22 × 5 database[104]) for Hartree−Fock[61,62] (HF), MP2, $\kappa$MP2 and MLS2. (**e, bottom**) Relative absolute correlation energy errors in log-scale of available systems from the W4-11 test dataset[102] (see Supplementary Section S12 in the SI for a detailed list of test data points) for MP2, $\kappa$MP2 and MLS2. The mean absolute relative errors (MArEs) are shown with dashed lines.

detailed in ref. 64, this ground distinguishes the MPAC-based correlation DFAs from density-corrected DFT, where HF densities are introduced heuristically to improve DFAs developed within KS DFT[75–77]. Leveraging this MPAC formalism and its recently introduced correlation energy densities[64], we construct DFAs as NN-based functionals of HF densities, enabling practical use of HF orbitals to compute all energy terms and input features for our NNs.

To briefly introduce our energy density per particle targets for LES, we define correlation energy as,

$$E_c = E^{\text{ref}} - \langle \phi | \hat{H} | \phi \rangle = \int e_c(\mathbf{r}) \rho(\mathbf{r}) d\mathbf{r} = \int \bar{e}_c(\mathbf{r}) d\mathbf{r}, \quad (4)$$

where $E^{\text{ref}}$ is the exact ground-state energy, $\hat{H}$ is the corresponding exact Hamiltonian, and $\Phi$ is the HF wavefunction (a single Slater determinant minimizer of $\hat{H}$, that yields $\rho^{\text{HF}}(\mathbf{r}) = \rho(\mathbf{r})$). In Eq. (4), we distinguish the correlation energy density, $\bar{e}_c(\mathbf{r}) = e_c(\mathbf{r})\rho(\mathbf{r})$, from the correlation energy per particle, $e_c(\mathbf{r})$, as this distinction is crucial for our subsequent LES analysis. As $e_c(\mathbf{r})$ is not uniquely defined, we adopt here a specific definition (i.e., gauge) for $e_c(\mathbf{r})$, derived in ref. 64 from the MPAC theory. This gauge is designed as an MPAC-based analogue[64] of the conventional DFT gauge for correlation energies, i.e., the electrostatic potential of the correlation hole[78–81], known for transparent physical interpretation and advantages in DFA construction[81,82]. Since computing the exact $e_c(\mathbf{r})$ is costly[64], we approximate it using its weakly interacting limit determined by second-order Møller-Plesset

perturbation theory[83] (MP2), preserving the original MPAC gauge,

$$e_c^{MP2}(\mathbf{r}) = \frac{1}{4\rho(\mathbf{r})} \int \frac{P_2^{MP2}(\mathbf{r},\mathbf{r}')}{|\mathbf{r}-\mathbf{r}'|} d\mathbf{r}', \tag{5}$$

where $P_2^{MP2}(\mathbf{r},\mathbf{r}')$ is the first-order MPAC correction to the pair density that yields the MP2 correlation energy (for its formal definition and derivation of Eq. (5), see Supplementary Section S1 in the SI). Crucially, $P_2^{MP2}(\mathbf{r},\mathbf{r}')$ isolates the correlation contribution to the pair density analogous to the DFT correlation hole[64].

Expressing Eq. (5) in terms of HF orbitals yields,

$$e_c^{MP2}(\mathbf{r}) = \frac{1}{\rho(\mathbf{r})} \left[ \sum_{ijab} V_{ijab}(\mathbf{r})\left(\tfrac{1}{2}T_{ijba} - T_{ijab}\right) \right. \\ \left. + \sum_{ijab} V_{ijba}(\mathbf{r})\left(\tfrac{1}{2}T_{ijab} - T_{ijba}\right) \right], \tag{6}$$

where $i, j$ are occupied, and $a, b$ are virtual KS orbital ($\phi(\mathbf{r})$) indices. $T_{ijab}$ are the partial MP2 doubles amplitude,

$$T_{ijab} = \frac{\langle ij|ab\rangle}{\varepsilon_a + \varepsilon_b - \varepsilon_i - \varepsilon_j}, \tag{7}$$

where $\varepsilon$ are orbital energies, and $V_{ijab}(\mathbf{r})$ is the orbital potential,

$$V_{ijab}(\mathbf{r}) = \phi_i(\mathbf{r})\phi_a(\mathbf{r}) \int \frac{\phi_j(\mathbf{r}')\phi_b(\mathbf{r}')}{|\mathbf{r}-\mathbf{r}'|} d\mathbf{r}'. \tag{8}$$

As $e_c^{MP2}(\mathbf{r})$ by Eq. (4) integrates to the MP2 correlation energy, it can be easily argued that it is not a sufficiently good proxy reference for LES-based applications given general MP2 limitations[68,69,84], particularly for small orbital-gap systems[85] (see the MP2 dissociation curve relevant to this work in Supplementary Fig. S1 in the SI). To address this, we apply Head-Gordon's $\kappa$-regularizaration[68,69] ($\kappa \geq 0$) to $e_c^{MP2}(\mathbf{r})$ of Eq. (5) by regularizing its partial MP2 doubles amplitudes,

$$T_{ijab}^{\kappa} = T_{ijab}\left(1 - e^{-\kappa(\varepsilon_a + \varepsilon_b - \varepsilon_i - \varepsilon_j)}\right)^2. \tag{9}$$

The regularized $e_c^{\kappa MP2}(\mathbf{r})$ is obtained from Eq. (6) by the replacement $T_{ijab} \to T_{ijab}^{\kappa}$, which also regularizes the underlying pair density (see Supplementary Section S1 in the SI):

$$e_c^{\kappa MP2}(\mathbf{r}) = \frac{1}{4\rho(\mathbf{r})} \int \frac{P_2^{\kappa MP2}(\mathbf{r},\mathbf{r}')}{|\mathbf{r}-\mathbf{r}'|} d\mathbf{r}'. \tag{10}$$

When $\kappa = 0$, $e_c^{\kappa MP2}(\mathbf{r}) = 0$, and when $\kappa \to \infty$, $e_c^{\kappa MP2}(\mathbf{r}) = e_c^{MP2}(\mathbf{r})$. In Fig. 1(d, top), we observe how $e_c^{\kappa MP2}(\mathbf{r})$ evolves to $e_c^{MP2}(\mathbf{r})$ for the helium dimer as $\kappa$ increases. Using the regularized $e_c^{\kappa MP2}(\mathbf{r})$ [blue square in Fig. 1(a)] instead of $e_c^{MP2}(\mathbf{r})$ [magenta circle in Fig. 1(a)] as the proxy reference for LES-based learning is crucial, since the former significantly improves dissociation curves of diatomic systems, which are central to our LES vs. GES comparison. In these curves, $\kappa$-regularization removes the MP2 divergence at large bond lengths arising from small orbital energy gaps. Here we set $\kappa = 2.0$ to generate $e_c^{\kappa MP2}(\mathbf{r})$ proxy reference, as it better balances improvements over MP2 for stretched bonds while maintaining accuracy near equilibrium, compared to the originally proposed $\kappa = 1.4$[68] (see Supplementary Fig. S1 in the SI for a clarifying example of $N_2$ dissociation).

After regularization, we perform the spin-resolution of $e_c^{\kappa MP2}(\mathbf{r})$ into same-spin (ss) and opposite-spin (os) components: $e_c^{\kappa MP2}(\mathbf{r}) = e_{c,ss}^{\kappa MP2}(\mathbf{r}) + e_{c,os}^{\kappa MP2}(\mathbf{r})$, enabling a spin-resolved real-space analysis of electron correlation [purple and orange in Fig. 1(a)]. The more compact of the two, $e_{c,os}^{\kappa MP2}(\mathbf{r})$, we derive from Eq. (6) by considering only os electronic pairs:

$$e_{c,os}^{\kappa MP2}(\mathbf{r}) = -\frac{1}{2\rho(\mathbf{r})} \sum_{ijab} \left[ T_{ijab}^{\kappa} V_{ijab}(\mathbf{r}) + T_{ijab}^{\kappa} V_{ijba}(\mathbf{r}) \right]. \tag{11}$$

Later, we will use ML for real-space scaling of $e_{c,os}^{\kappa MP2}(\mathbf{r})$ and $e_{c,ss}^{\kappa MP2}(\mathbf{r})$ separately to build a correction from $\kappa$MP2 to the true $E_c$.

Central to our data generator for real-space ML of $e_c(\mathbf{r})$ are the $\kappa$-regularization and spin resolution of MP2 correlation energy densities, both efficiently implemented by combining density fitting[86,87] with the power of JAX[70] and modern tensor libraries[88] for optimizing tensor contractions required to obtain $e_c^{\kappa MP2}(\mathbf{r})$. Further implementation details are provided in Methods and Supplementary Section S3 of the SI.

## Regularized PT2 correlation energy densities for interaction energies

Now we move to a real-space analysis of the interaction correlation energy density, $\Delta\bar{e}_c(\mathbf{r})$, defined as the total system's $\bar{e}_c(\mathbf{r})$ minus the sum of the $\bar{e}_c(\mathbf{r})$ of the isolated subsystems (e.g., dimer minus monomers). A subtle point here is that while for ML purposes $e_c(\mathbf{r})$ is vastly superior to $\bar{e}_c(\mathbf{r})$ (see below), spatial visualization of interactions requires $\Delta\bar{e}_c(\mathbf{r})$ instead of $\Delta e_c(\mathbf{r})$, as the former directly integrates to $\Delta E_c$, while the latter lacks a clear density factor to do so.

From Eq. (10), $\Delta\bar{e}_c^{\kappa MP2}(\mathbf{r})$ is given by the electrostatic potential of $\Delta P_2^{\kappa MP2}(\mathbf{r},\mathbf{r}')$,

$$\Delta\bar{e}_c^{\kappa MP2}(\mathbf{r}) = \frac{1}{4} \int \frac{\Delta P_2^{\kappa MP2}(\mathbf{r},\mathbf{r}')}{|\mathbf{r}-\mathbf{r}'|} d\mathbf{r}', \tag{12}$$

where $\Delta P_2^{\kappa MP2}(\mathbf{r},\mathbf{r}')$ is the interaction component of $P_2^{\kappa MP2}(\mathbf{r},\mathbf{r}')$. As $P_2^{\kappa MP2}(\mathbf{r},\mathbf{r}')$ isolates the correlation part of the underlying pair density, $\Delta P_2^{\kappa MP2}(\mathbf{r},\mathbf{r}')$ further isolates how this quantity is deformed by the interaction between fragments, making this quantity crucial for describing the physics of weak interactions, particularly dispersion effects (see, e.g., refs. 89–91).

While Eq. (12) represents just one possible gauge for $\Delta\bar{e}_c^{\kappa MP2}(\mathbf{r})$, we emphasize that it encodes the physics of weak interactions through its direct link to $\Delta P_2^{\kappa MP2}(\mathbf{r},\mathbf{r}')$, a fundamental quantity for describing the physics of weak interactions, such as dispersion effects[89–91]). To illustrate this for dispersion, we first consider a simple example: the helium dimer, for which spin-resolved interaction correlation energy densities $\Delta\bar{e}_c^{MP2}(\mathbf{r})$ are shown in Fig. 1 (d, bottom) along the internuclear axis. We now show their sum in Fig. 2(a), highlighting distinct binding (negative) and non-binding (positive) contributions for stretched $He_2$. To illustrate the interaction physics encoded in $\Delta\bar{e}_c^{MP2}(\mathbf{r})$, we fix the reference electron at the position labeled by $z_0$ in $He_2$, and in the inset of Fig. 2(a) we show $\Delta P_2^{MP2}(z_0, z')$ along the internuclear axis as a function of the second electron position $z'$. The plot reveals spatially nonlocal polarization in $\Delta P_2^{MP2}$, characteristic of dispersion: when $z'$ is near the second helium nucleus, $\Delta P_2^{MP2}(z_0, z')$ exhibits a negative accumulation closer to $z_0$ and a corresponding positive region further away. As the negative part of $\Delta P_2^{MP2}(z_0, z')$ lies closer to $z_0$ than the positive part, the resulting electrostatic potential, i.e., $\Delta\bar{e}_c(z_0)$, is negative. If the polarization pattern is reversed, i.e., the positive part of $\Delta P_2^{MP2}(z_0, z')$ lies closer to $z_0$ than the negative part, as when $z_0$ is in the outer region, then $\Delta\bar{e}_c(z_0)$ becomes positive (see Supplementary Fig. S3 for additional plots in the SI). Still, the negative (i.e., binding) regions dominate, and MP2 correlation yields net binding in $He_2$ (Supplementary Table S1 in the SI). In this way, Eq. (12) condenses information from $\Delta P_2(\mathbf{r},\mathbf{r}')$, a key two-body quantity encoding interaction physics, into $\Delta\bar{e}_c(\mathbf{r})$, a one-body correlation quantity for interaction energies. Thus, even though it is not unique, the specific gauge defined by Eq. (12) yields a physically interpretable local correlation energy quantity for describing weak interactions between fragments.

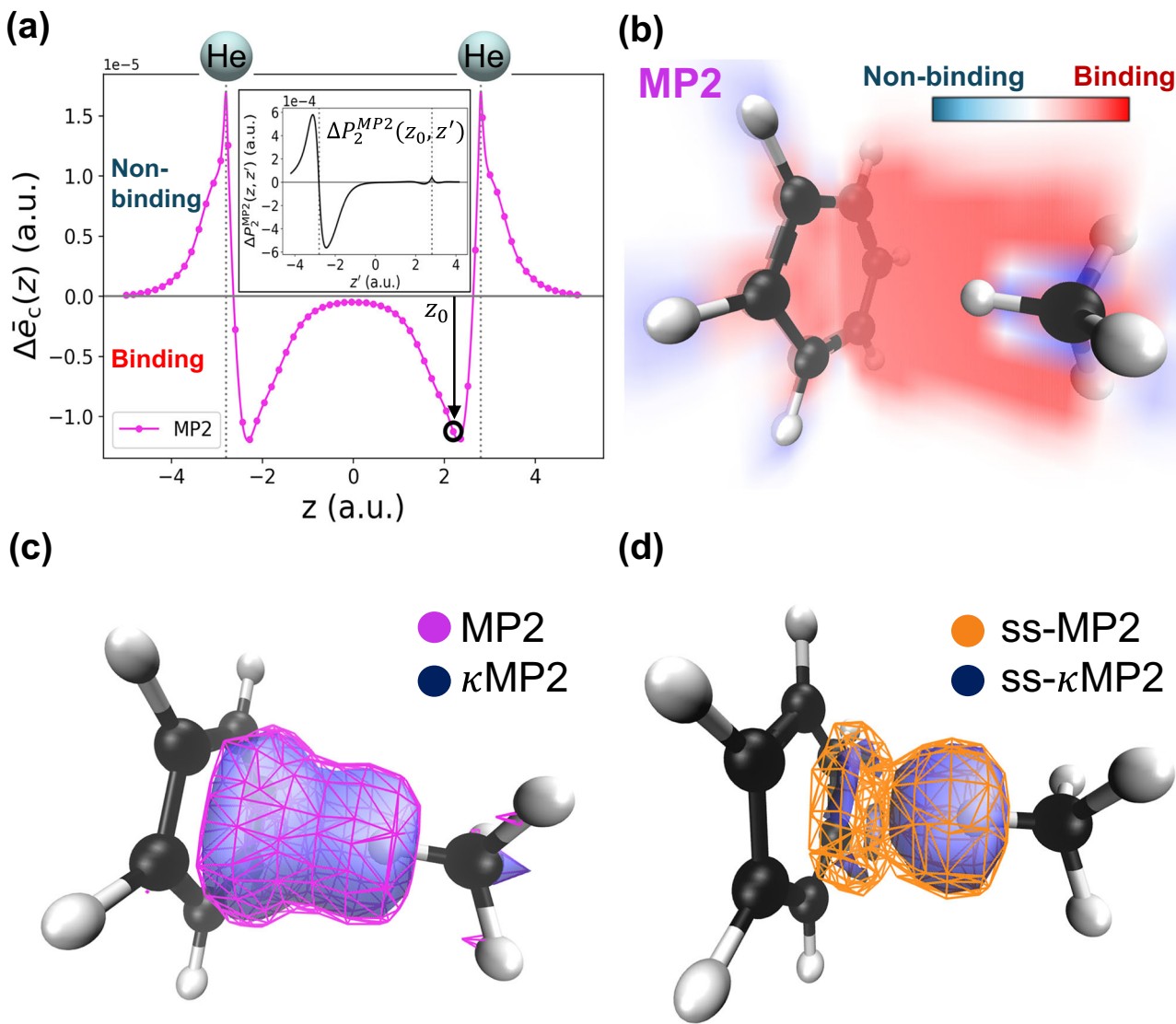

**Fig. 2 | Visualization of $\Delta \bar{e}_c(\mathbf{r})$ (total correlation energy density minus the one from the individual subsystems). a** Second-order Møller−Plesset perturbation theory (MP2) plot along the inter-nuclear axis of the helium dimer (position of the nuclei are denoted by the spheres) at an interatomic distance of 5.6 a.u.. Inset shows the corresponding interaction pair density $\Delta P_2^{MP2}(\mathbf{r}, \mathbf{r}')$ at $\mathbf{r} = z_0$ on the same axis. **b** MP2 volume slice plots for the benzene-CH$_4$ complex[85] along planes perpendicular and parallel to the benzene ring. Binding and non-binding regions are highlighted by negative (red) and positive (blue) $\Delta \bar{e}_c(\mathbf{r})$ volumes. **c** Isosurface visualization of MP2 (rough mesh) and its $\kappa$-regularization ($\kappa$MP2) with $\kappa = 1.4$ (solid surface) for the same complex at binding isovalue ($-2.5\text{e}{-5}$ (a.u.)). **d** Same as (**c**), but for the ss part.

Having established the direct connection between $\Delta \bar{e}_c^{MP2}(\mathbf{r})$ and dispersion physics, we now analyze $\Delta \bar{e}_c^{MP2}(\mathbf{r})$ for the benzene-methane complex in the remaining panels of Fig. 2. In Fig. 2(b), volume slices along planes perpendicular and parallel to the benzene ring distinguish regions between the fragments (typically binding regions) from those outside (typically non-binding). MP2 overbinds this complex, whereas $\kappa$MP2 reduces this overbinding (Supplementary Table S2 in the SI). This reduction is visually reflected in Fig. 2(c), which compares MP2 and $\kappa$MP2 $\Delta \bar{e}_c(\mathbf{r})$ isosurfaces for the binding region, with the $\kappa$MP2 isosurface confined within the MP2 counterpart. The spatial confinement is even more pronounced when focusing on just the ss component of $\Delta \bar{e}_c^{\kappa MP2}(\mathbf{r})$ [the electrostatic potential of the ss component of the underlying $\Delta P_2^{\kappa MP2}(\mathbf{r}, \mathbf{r}')$], as shown in Fig. 2(d).

**ML2 model via machine-learning of regularized $e_c^{\kappa MP2}(\mathbf{r})$**

We will now present the ML2 model, a machine-learned correlation energy density based on the regularized MP2 proxy reference, which we obtain by mapping a set of pointwise features using neural networks (NNs) [going from the blue square to the cyan or maroon diamonds in Fig. 1(a)]. This mapping is illustrated in Fig. 1(b). In addition to the established features used in ML of DFAs[31,56]−the reduced density gradient $s(\mathbf{r})$, the reduced density Laplacian $q(\mathbf{r})$, and the regularized kinetic energy variable $\alpha(\mathbf{r})$ from the $r^2$SCAN DFA[92]−we introduce Grimme's electronic temperature-dependent fractional occupation number weighted density[71,72] (FOD) as the crucial feature (a detailed list of features is given in Supplementary Section S6 in the SI). FOD differentiates strongly correlated and weakly correlated regions in molecules. While both $e_c^{\kappa MP2}(\mathbf{r})$ and FOD provide insights into electronic correlation through the interaction between occupied and unoccupied orbitals, FOD is computationally much cheaper, making it an excellent feature for ML of the correlation energy density.

Even though $\rho(\mathbf{r})$-based features (e.g., the Wigner−Seitz radius) might seem like a natural choice, we intentionally omit them from the ML2 features. This is because we found that $e_c^{MP2}(\mathbf{r})$ and its

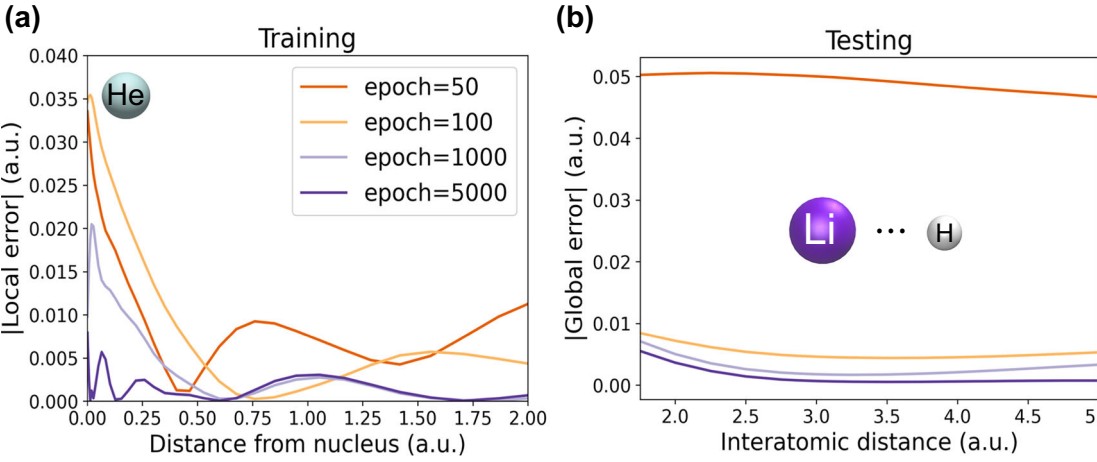

**Fig. 3 | Errors of Local Energy Loss (LES) based machine-learned energy densities from regularized second-order Møller–Plesset perturbation theory (ML2) at different learning epochs. a** Absolute (local) error in correlation energy densities per particle for the helium atom. **b** Absolute global energy error for LiH along its dissociation curve.

$\kappa$-counterpart are scaling invariant for a uniformly scaled[93] density $\rho_\lambda(\mathbf{r}) = \lambda^3 \rho(\lambda\mathbf{r})$,

$$e_c^{\kappa MP2}[\rho_\lambda](\mathbf{r}) = e_c^{\kappa MP2}[\rho](\mathbf{r}). \tag{13}$$

A detailed derivation is given in Supplementary Section S7 in the SI. Following this scaling invariance, we construct $e_c^{ML2}(\mathbf{r})$, the ML2 analog of $e_c^{\kappa MP2}(\mathbf{r})$, as

$$e_c^{ML2}(\mathbf{r}) = w_c(\mathbf{r})\, e_x(\mathbf{r})\, \rho^{-1/3}(\mathbf{r}), \tag{14}$$

where $e_x(\mathbf{r})$ is the exchange energy (we use the same Python code to implement both $e_c^{MP2}(\mathbf{r})$ and $e_x(\mathbf{r})$ on the same footing), and $w_c(\mathbf{r})$ are the ML2 weights obtained from the NN (see Fig. 1(b) for the illustration and Methods for NN architecture details). With the use of HF-based ingredients, the computational cost of ML2 is comparable to that of an HF calculation and cannot be lower. As we shall see later, embedding the physics into LES-based ML2 through Eq. (14) is crucial for the robustness of the model.

Using the same features, functional form of Eq. (14), and NN architecture for mapping the features at a given $\mathbf{r}$ to $w_c(\mathbf{r})$, we can now isolate the difference between using LES and GES for NN training of a DFA [cyan vs. maroon diamonds in Fig. 1(a)]. For simplicity and to create a challenging transferability test, we train our ML2-based NN only on eight small closed-shell atoms/ions (H⁻, He, Be, Ne, Mg, Ar, Ca, and Kr). The total loss is calculated as the mean over these eight datapoints, as detailed in Supplementary Section S8 in the SI. We validate our training on comparable small closed-shell atoms/ions (see Supplementary Fig. S4) in the SI.

In Fig. 1(c), we explore GES-based vs. LES-based ML2 results (cyan vs. maroon diamonds). Both energy densities are plotted against the $e_c^{\kappa MP2}(\mathbf{r})$ reference in Fig. 1(c, top) for the Mg atom as one of the training datapoints. We can see that $e_c^{ML2}(\mathbf{r})$ based on LES closely follows the $e_c^{\kappa MP2}(\mathbf{r})$ (proxy) reference, whereas the GES-based $e_c^{ML2}(\mathbf{r})$ completely misses the shape of the reference. But, can we, solely based on this observation, conclude that LES is better than GES? We need to be careful here, as these correlation energy densities are not observables even within our physically sound gauge (see previous section). Instead, they are used here to enhance energy training data efficiency in ML of DFAs, expanding from one to thousands of energy datapoints per system. Thus, what ultimately matters for judging the quality of GES vs. LES training are the integrated correlation energies (Eq. (1)) once we go outside of the training set.

For the Mg case in Fig. 1(c, top), all energy densities integrate to nearly the same correlation energy by Eq. (1). However, the correlation energies from our GES-based and LES-based ML2 models differ dramatically when tested for transferability. The first such test is shown in Fig. 1(c, bottom), where we see major differences in accuracy when applied to stretching the BH diatomic system along its dissociation curve. The LES-based model closely follows the $e_c^{\kappa MP2}(\mathbf{r})$ (proxy) reference result, while the GES-based model is not sufficiently accurate even at equilibrium and breaks down completely as the bond is stretched.

## The advantages of LES-based learning process
After seeing in Fig. 1(c, bottom) that LES-based ML2 trained only on atomic systems successfully transfers to the BH diatomic along its dissociation curve, we now investigate this atom to diatomics transferability in more detail. Namely, a closer look at the learning process of the LES-based ML2 is given in Fig. 3, where panel (a) shows the absolute error in $e_c^{ML2}(\mathbf{r})$ for the He atom (one of ML2's training points) at different epochs. On the other hand, panel (b) focuses on the test and shows the absolute error in the ML2 correlation energy along the LiH dissociation curve for the same set of epochs. Overall, Fig. 3(a) shows that the pointwise improvement in $e_c^{ML2}(\mathbf{r})$ for the He atom during training translates epoch by epoch into improved correlation energies for the unseen LiH system, as indicated by the decreasing errors with increasing epochs in panel (b). In contrast to that, the GES-based model yields no improvements of the transferability from atoms as the learning process progresses (see Supplementary Fig. S5 in the SI exposing poor transferability to diatomics, especially at large distances even at large epochs). Furthermore, we observe another crucial difference between GES-based and LES-based learning: when LES is used as the loss function, the learning process is much smoother and it has a much faster convergence with respect to learning steps (epochs) compared to GES (see Supplementary Fig. S6 in the SI).

Finishing off, by making energies more information-rich through LES in the training, we equip our models with a high level of transferability, ensuring that any information learned from atoms lead to better performance on molecules. LES reveals information completely washed away with GES, showing that combinations of features and corresponding energy densities per particle, even for atoms, are highly relevant for molecules. Thus, LES provides a powerful strategy for ML of transferable DFAs, and in what follows we will explore more subtle details linked to the LES-based training.

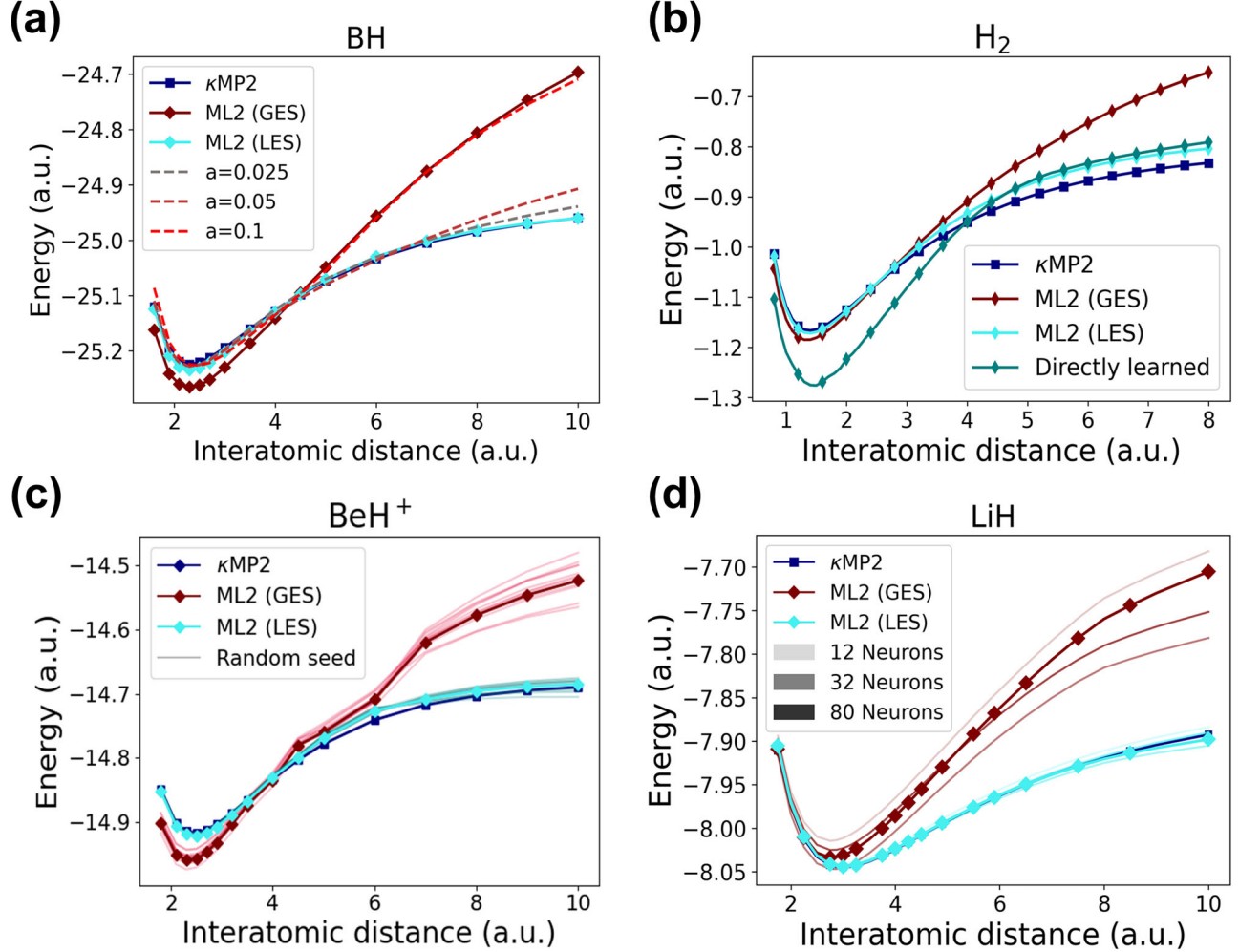

**Fig. 4 | Dissociation curves as in Fig. 1(c, bottom), but for four different systems with additional data for comparison.** Shown are machine-learned results (ML2) from $\kappa$-regularized second-order Møller-Plesset perturbation theory ($\kappa$MP2) using Global Energy Loss (GES) or Local Energy Loss (LES). **a** BH curves including ML2 results that employ LES with proxy correlation energy densities coming from different $a$ – parameter dependent gauges (see Eq. (15)). **b** $H_2$ curves including 'directly

learned' model that learns $\bar{e}_c(\mathbf{r})$ from $\kappa$MP2 based on a direct loss (see text). **c** $BeH^+$ curves including GES- and LES-based ML2 results coming from different random seed initializations. **d** LiH curves including additional ML2 results coming from neural networks with different total number of neurons (see Supplementary Fig. S11 in the SI for the distribution of neurons per hidden layer).

## Uniqueness and robustness of our LES-based ML2 model

Building on the successful LES-based ML2 transferability from atoms to challenging BH and LiH dissociations, Fig. 4 shows dissociation curves for four additional diatomics, further confirming the LES's advantage over GES, which can yield unphysical curves. The dissociation curves for closed-shell diatomics are obtained using a spin-restricted formalism to avoid artificial energy lowering from spin-symmetry breaking (see refs. 94–96 for discussions on the challenge of describing bond breaking without spin-symmetry breaking). We now present four analyses, each applied to a different curve from Fig. 4, to highlight distinct aspects of LES-based ML2 transferability. Importantly, the conclusions from each analysis are robust, consistently holding when cross-checked against other diatomics, as demonstrated in the SI (Supplementary Figs. S7–S11).

We first test whether the superior transferability of LES over GES from atoms to diatomics arises purely from LES's higher data efficiency. Specifically, panels (a) and (b) of Fig. 4 show that this transferability is lost if the LES-based ML2 model is constructed without the physics encoded in Eq. (3) (LES defined via energy densities per particle), Eq. (10) (specific $e_c(\mathbf{r})$ gauge), and Eq. (14) (physically constrained ML2 functional form).

To examine the sensitivity of ML2 training to the gauge choice of $e_c^{\kappa MP2}(\mathbf{r})$ (Eq. (5)), we introduce the following gauge transformation:

$$e_c^a(\mathbf{r}) = e_c^{\kappa MP2}(\mathbf{r}) + a\, q(\mathbf{r})\rho^{2/3}(\mathbf{r}), \qquad (15)$$

with $q(\mathbf{r}) = [\nabla^2\rho(\mathbf{r})]/[4(3\pi^2)^{2/3}\rho(\mathbf{r})^{5/3}]$, and where the real parameter $a$ does not affect the integrated correlation energy (Eq. (4)) for exponentially decaying densities. Instead of our original target, $e_c^{\kappa MP2}(\mathbf{r})$, we now repeat the LES-based ML2 training using $e_c^a(\mathbf{r})$ as the target at various $a$ values. In Fig. 4(a), we test the transferability of the underlying $a$-dependent model by using the BH dissociation curve previously shown in Fig. 1(c, bottom). These curves become worse as we move away from $a = 0$ (original LES-based ML2) and even become unphysical at larger $a$ (the results for other diatomic dissociation curves follow similar trends as shown in Supplementary Fig. S7 in the SI). Observing $e_c^a(\mathbf{r})$ for atoms (train) and BH (test) in Fig. S8 in the SI, we see that at $a = 0$, the range and shape of $e_c^a(\mathbf{r})$ varies much less between train and test systems compared to larger $a$ values, explaining why the original LES-based ML2 ($a = 0$) exhibits the best transferability. While Eq. (15) does not cover all possible gauges, our tests show that the transferability from atoms to diatomics achieved by our gauge is

highly nontrivial, and that superior performance of LES over GES in our ML2 model is not solely due to higher data efficiency but critically depends on the gauge choice (Eq. (10)) for the $e_c^{ML}(\mathbf{r})$ training target (observe how $\kappa$MP2 is more amenable for ML in Supplementary Fig. S8 in the SI).

To further demonstrate that data efficiency alone is insufficient for ML2's success, we compare dissociation curves for $H_2$ from various models in Fig. 4(b). The figure highlights the importance of defining LES in terms of energy densities per particle (Eq. (3)) and employing the physically constrained LES-based ML2 form (Eq. (14)). If, instead of LES, the loss is defined directly via energy densities (not per particle)[97–99], $\mathcal{L}_{\text{direct}} \sim \int |\bar{e}_c^{\text{ref}}(\mathbf{r}) - \bar{e}_c^{ML}(\mathbf{r})| d\mathbf{r}$, then the resulting model ("directly learned") performs as bad as the GES-based model (see Fig. 4(b)). Additional examples of even more drastic failures of $\mathcal{L}_{\text{direct}}$-based learning, illustrating the subtle yet crucial importance of learning $e_c(\mathbf{r})$ rather than $\bar{e}_c(\mathbf{r})$, are shown in Supplementary Fig. S9 in the SI. The poor model's transferability when direct loss is used (learning $\bar{e}_c^{ML}(\mathbf{r})$) in place of LES (learning $e_c^{ML}(\mathbf{r})$) is unsurprising, given that both $e_c(\mathbf{r})$ and the ML2 weights, $w_c(\mathbf{r}) = e_c(\mathbf{r})/(e_x(\mathbf{r})\rho^{-1/3}(\mathbf{r}))$ (Eq. (14)), are far less sensitive to variations in system size than $\bar{e}_c(\mathbf{r}) = e_c(\mathbf{r})\rho(\mathbf{r})$. This comparison further emphasizes that LES-based ML2's transferability success does not rely solely on higher data efficiency than GES, but critically depends on the synergy between this efficiency and the physics embedded in Eqs. (3), (10), and (14).

Finally, we demonstrate the robustness of LES-based ML2 with respect to NN training conditions: unlike GES, LES-based ML2 remains stable under variations in random initialization seeds (Fig. 4(c) for BeH⁺; additional examples in Supplementary Fig. S10 in the SI) and NN architecture, including the number of neurons (Fig. 4(d) for LiH; additional examples in Supplementary Fig. S11 in the SI). We also show in Supplementary Fig. S12 in the SI that the use of mean square-based LES (i.e. LES² instead of the original absolute differences-based LES of Eq. (3)) has little effect on the ML2 results. While one may argue that training on only eight atomic systems leaves the GES model prone to overfitting and poor generalization, the good performance of LES with the same small number of systems is already its core advantage over GES in the low-data regime. Nevertheless, we also show that as the training set is progressively enlarged, LES remains more robust than GES for diatomic dissociation curves, even when additional diatomic systems are included in the GES training (see Supplementary Section S10 and Supplementary Fig. S13 in the SI). Overall, the robustness and uniqueness of the LES-based ML2 model demonstrated in this section further emphasize the advantages of LES-based ML2.

## Spin-resolved and regularized modeling of the correlation energy density

In the previous section we have shown that LES enhances the transferability of ML DFAs. Yet, $\kappa$MP2 correlation has been the proxy reference in place of its exact counterpart. In this section, we use our regularized PT2-based generator for a NN-based combination of the spin-resolved $\kappa$MP2 energy densities to bridge the gap between $\kappa$MP2 and true correlation energies. For this purpose, our ML model for correlation energy densities per particle is defined as

$$e_c^{MLS2}(\mathbf{r}) = w_{os}(\mathbf{r})e_{c,os}^{\kappa MP2}(\mathbf{r}) + w_{ss}(\mathbf{r})e_{c,ss}^{\kappa MP2}(\mathbf{r}),\qquad(16)$$

where $w_{os}(\mathbf{r})$ and $w_{ss}(\mathbf{r})$ are machine-learned weights at every point in space. We call it MLS2, which represents a real-space, machine-learned and regularized extension of SCS MP2[73,74]. Its construction is represented by the step from orange and purple squares to the green diamond in Fig. 1(a). By leveraging our implementation of spin-resolved $w_{os}(\mathbf{r})e_{c,os}^{\kappa MP2}(\mathbf{r})$ and $w_{ss}(\mathbf{r})e_{c,ss}^{\kappa MP2}(\mathbf{r})$, Eq. (16) yields a regularized and real-space extension of SCS MP2, thus opening up avenues for DFAs creation.

To obtain $w_{os}(\mathbf{r})$ and $w_{ss}(\mathbf{r})$ in MLS2 using a NN, we employ a similar architecture as for the ML2 model (see Fig. 1(b)) with some crucial differences. First, unlike the MP2 correlation energy, the true correlation energy is generally not scale invariant[93]. This allows us contrary to ML2 (see Eq. (13) and the preceding paragraph) to incorporate also density-based features, specifically, the Wigner-Seitz radius. Second, a sigmoid activation function is applied to the NN's output layer (more details on the activation functions between layers are given in Methods), constraining the MLS2 weights between 0 and 1. We scale the resulting NN weights $w_{os}(\mathbf{r})$ and $w_{ss}(\mathbf{r})$ by a constant factor of 10 before applying them in Eq. (16). This scaling enables the MLS2 model to be accurate for the cases where the true correlation energy (in absolute terms) is much larger than what $\kappa$MP2 predicts (see Supplementary Section S11 in the SI for more details). Finally, we use $e_c^{\kappa MP2}(\mathbf{r})$ and $e_{c,os}^{\kappa MP2}(\mathbf{r})$ quantities normalized by $\rho^{-1/3}(\mathbf{r})e_x(\mathbf{r})$ as extra MLS2 dimensionless features (see Supplementary Section S6 in the SI for a detailed list of features).

Ideally, with access to a robust data generator for the exact $e_c(\mathbf{r})$, we could use LES to train $e_c^{MLS2}(\mathbf{r})$. However, due to the severe computational limitations of such a generator[80,100] to very small systems and basis sets, we settle with a GES-based training of MLS2 (the resulting loss function is detailed in Supplementary Section S8 in the SI). Using GES here requires more data for training. Thus, we employ the eight atoms/ions training dataset from ML2 combined with 13 small closed-shell molecules, mainly dimers, and correlation energies of $H_2$, $N_2$ and $Li_2$ at five geometries of large interatomic distances. In addition to these total energies, our MLS2 NN is also trained on interaction energies from the RG18 dataset[101], including dispersion-bound complexes with noble gases. We elaborate in Supplementary Section S8 of the SI on how we combine the total energy-based GES and the interaction energy-based GES when training MLS2. Furthermore, a detailed MLS2's list of all training data points is given in Supplementary Section S12 of the SI.

To test MLS2, we go back to Fig. 1(e, bottom), which includes results for 96 total correlation energies from the W4-11 database[102] not present in the training set (see Supplementary Section S12 in the SI for a full list and how we obtain the underlying reference total correlation energies). Specifically, Fig. 1(e, bottom) shows the relative absolute correlation energy errors for $\kappa$MP2, MP2 and MLS2. Note the log-scale in the y-axis and the dashed lines representing mean absolute relative errors (MArEs). Going from MP2 to $\kappa$MP2 (from magenta circles to blue squares), we can see that the introduction of the $\kappa = 2.0$ regularization slightly increases the MP2 errors. On the other hand, our MLS2 model (green diamonds) yields here far more accurate correlation energies than MP2, with MArE reduced from ~10% to 1%.

In Fig. 5, we revisit the dissociation curve of BH to test MLS2 as the bond stretches and include additional methods beyond those shown in Fig. 1(c, bottom). Unsurprisingly, MP2 (magenta curve) fails to capture the correct physics of BH bond stretching due to the divergence of its correlation energies when the HOMO-LUMO orbital gap closes (see Eq. (7)). $\kappa$MP2 (blue curve; Eq. (9)) eliminates this divergence, but its energies are much higher than the exact ones when the bond stretches. In contrast, MLS2 is more accurate than $\kappa$MP2 not only when the unseen BH bond stretches but also at equilibrium (see also Supplementary Fig. S15 in the SI for training results of $N_2$ and $H_2$ bond stretches). This demonstrates the power of MLS2 to successfully dissociate covalent bonds without breaking spin symmetries, which is, as said, a crucial challenge for quantum chemistry methods[94–96,103]. In MLS2, this is achieved by first employing $\kappa$MP2 to eliminate the divergence present in MP2, followed by real-space NN-based enhancements of its spin-resolved energy densities (see Eq. (16)).

Finally, revisiting Fig. 1(e, top) with the formic acid dimer interaction energy curve, we test the performance of MLS2 for hydrogen-bonded systems. The formic acid dimer is selected because of its two

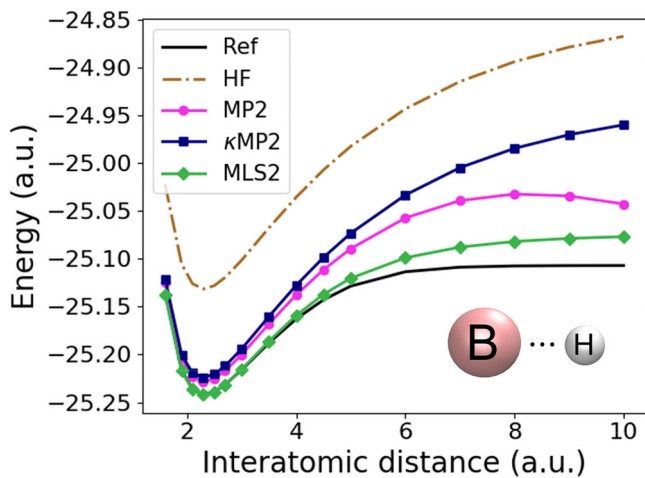

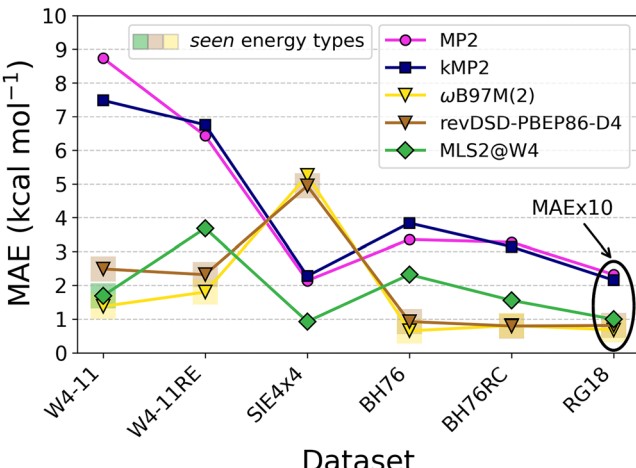

**Fig. 5 | Test result of real-space, machine-learned, and regularized extension of spin-component-scaled second-order Møller–Plesset perturbation theory (MLS2).** Dissociation energy curve of the BH diatomic system, as in Fig. 1(c, bottom), but with additional models (reference values (Ref) taken from ref. 124, Hartree–Fock (HF), second-order Møller–Plesset perturbation theory (MP2) and its $\kappa$-regularized version ($\kappa$MP2)).

**Fig. 6 | Energy mean absolute error (MAE) in kcal mol⁻¹ of various models for subsets of the GMTKN55 database[105] and the additional W4-11RE set[106].** The real-space, machine-learned, and regularized extension of spin-component-scaled second-order Møller-Plesset perturbation theory (MLS2) model is trained on W4-11 atomization energies[102] (MLS2@W4). MAEs of second-order Møller-Plesset perturbation theory (MP2), its $\kappa$-regularization ($\kappa$MP2) and of two double-hybrid models ($\omega$B97M(2)[108] and revDSD-PBEP86-D4[109]) are shown for comparison. Translucent squares behind the markers denote seen energy types for the corresponding models (see Methods). The results for RG18 dataset[101] are scaled by a factor of 10 for better visibility.

hydrogen bonds with a strong electrostatic component, making it a system that starkly contrasts the dispersion-bonded RG18 systems on which MLS2 was trained. Overall, MP2 overbinds the formic acid dimer, with $\kappa$MP2 overbinding even more, while MLS2 outperforms both in predicting the dimer's interaction energies.

We observe similar improvements of MLS2 over $\kappa$MP2 for other dissociation curves of noncovalent systems from the S22 × 5 dataset[104] (see Supplementary Fig. S16 in the SI). These results further confirm MLS2's transferability, as it successfully extrapolates from dispersion-bound systems (RG18) to distinctly different noncovalent interactions, such as hydrogen bonds.

To go beyond the closed-shell systems considered so far and demonstrate MLS2's broader applicability, we retrain it on the W4-11 atomization energy dataset[102] and test its performance on representative subsets of GMTKN55 (a large main-group database)[105]. We denote this variant as MLS2@W4. Since the MLS2@W4 training and test sets include open-shell systems, we supplement the existing MLS2 features (Supplementary Section S6 in the SI) with the spin polarization function $\zeta(\mathbf{r})$ (Supplementary Eq. (S12) in the SI). In Fig. 6, we summarize MLS2@W4 performance with the MAEs testing its transferability from W4-atomizations[102] to unseen energy types (see Supplementary Section S13 in the SI for further details): reaction energies (W4-11RE) with ~ 11k reaction data points derived from the W4-11 total energies[106], dissociation energies of small open-shell cationic dimers (SIE4 × 4)[107], barrier heights (BH76) and reaction energies (BH76RC)[105], and noncovalent interaction energies[105] (RG18). Results for MP2, $\kappa$MP2, and state-of-the-art double-hybrid DFAs ($\omega$B97M(2)[108] and revDSD-PBEP86-D4[109]) are included for comparison. First, Fig. 6 shows that MLS2@W4 clearly outperforms MP2 and $\kappa$MP2, demonstrating strong transferability from atomization energies alone to other energy types, even though it relies on a neural network with over 10k trainable parameters. In Fig. 6, translucent squares mark seen energy types (see Methods). While MLS2@W4 has only seen atomization energies, the two double hybrids have seen nearly all of the energy types shown, making the distinction between seen and unseen crucial for interpreting these results. Second, although the two double hybrids perform better for barrier heights and reaction energies (energy types seen by them and unseen by MLS2@W4), MLS2@W4 still achieves reasonable accuracy for these sets. For the SIE4 × 4 dataset, which is very difficult for the standard DFAs due to self-interaction errors[105],

with MAE slightly below 1 kcal mol⁻¹, MLS2@W4 impressively outperforms both double hybrids by a factor of ~ 5. Overall, MLS2@W4 stands out as the most robust method considered here, being the only one that achieves MAE below 4 kcal mol⁻¹ across all tested datasets. This further confirms that the general MLS2 framework combining ML with Eq. (16) is highly promising for developing future DFAs.

In view of the good MLS2 performance, it is important to note the regularizing role of Eq. (16). In addition to using a full amount of exact exchange (correlation here modeled relative to the HF reference), Eq. (16) ensures that MLS2 is exact for one-electron systems ($E_c^{\mathrm{MLS2}} = 0$). This is because, $e_{c}^{\mathrm{MLS2}}(\mathbf{r}) = e_{c,ss}^{\kappa\mathrm{MP2}}(\mathbf{r}) = e_{c,os}^{\kappa\mathrm{MP2}}(\mathbf{r}) = 0$ for $N = 1$, irrespective of the weights produced by the NN. This good property of MLS2 and likely its good transferability would be easily lost if other terms (e.g., exchange energy density or semilocal quantities multiplied by corresponding ML weights) are added to Eq. (16). Crucially, through the specific use of Eq. (16) and NNs, MLS2 provides a way of employing semilocal features [e.g., $s(\mathbf{r})$] while still satisfying the one-electron constraint, in contrast to double hybrids, which violate this constraint due to their way of employing semilocal features. Consequently, MLS2@W4 is not only exact for one-electron systems such as $H_2^+$ (by contrast, the two double hybrids yield substantial errors upon stretching $H_2^+$ as shown in Supplementary Fig. S17 in the SI), but also performs well for self-interaction cases involving more electrons given its excellent performance on SIE4 × 4.

## Discussion

To address the urgent need for transferable ML DFAs, we introduce and analyze several key strategies based on real-space energy learning. By leveraging our regularized, spin-resolved PT2-based correlation energy density generator, we pursue two directions, each demonstrating a distinct aspect of the power of real-space ML for DFAs.

### ML2 demonstrates the power of LES

The first direction, ML2, leverages $e_c^{\kappa\mathrm{MP2}}(\mathbf{r})$ (Eq. (5)) as a proxy reference for LES-based learning. While LES intrinsically expands a single energy

datapoint of GES into thousands per molecule (each grid point becoming a distinct training datapoint), this data efficiency advantage is fully realized only when specific physical considerations are accounted for. These include the use of energy density per particle as the learning target (Eq. (3)), adopting the physically meaningful gauge (Eq. (5)), and employing a physically-informed ML model (Eq. (14)). When these conditions are met, LES provides significantly greater transferability compared to commonly used GES. In particular, our ML2 model trained solely on a small set of atoms effectively generalizes to diatomic dissociation curves. Moreover, under these physically-informed conditions, LES not only enhances transferability but also leads to smoother and faster learning convergence, as well as robustness with respect to variations in ML training conditions compared to GES.

## MLS2 leverages our local quantities to develop transferable ML DFAs

In the second direction, MLS2, we employ our $e_c^{\kappa MP2}(\mathbf{r})$ and its decomposition into same-spin and opposite-spin channels to construct a real-space ML model for transferable DFAs. Specifically, MLS2 generalizes spin-component-scaled MP2 by scaling each spin channel locally with NN weights (Eq. (16)), combined with Head-Gordon's $\kappa$-regularization. MLS2 improves over $\kappa$MP2 across diverse systems, achieves competitive accuracy compared to modern double hybrids, and outperforms them for challenging systems affected by self-interaction errors. While the MLS2 ingredients [Eq. (16)] scale as their global ($\kappa$MP2) counterparts, further algorithmic developments (e.g., ref. 110) will be required to reduce the cost to a level comparable with $\kappa$MP2. Alternatively, ML2 surrogates of the MLS2 ingredients could be designed to lower the cost (see text below).

Going back to the demonstrated power of LES over GES in the proof-of-principle ML2 model based on the $\kappa$MP2 reference calls for developing robust energy density generators at higher levels of theory. This will be a crucial objective in our future work, with the first step already taken in ref. 64, which enables obtaining $e_c(\mathbf{r})$ from full configuration interaction (FCI) wavefunctions, with the procedure being easily adaptable to other variational wavefunctions. To learn higher-level $e_c(\mathbf{r})$, one can adapt the LES-based ML2 model, currently designed for $e_c^{\kappa MP2}(\mathbf{r})$, by adding $r_s$ in the features list and modifying Eq. (14). Nevertheless, we believe that a more controlled approach for ML of higher-level $e_c(\mathbf{r})$ can be achieved by integrating ML2 and MLS2 as follows. Using higher-level energy densities per particle, we can implement LES-based training for MLS2, while simultaneously replacing $e_{c,os}^{\kappa MP2}(\mathbf{r})$ and $e_{c,ss}^{\kappa MP2}(\mathbf{r})$ of Eq. (16) with their ML2-based surrogates. This replacement avoids these two more expensive quantities in post-training calculations while leveraging existing ML2 physics (Eq. (14)).

Within the MPAC framework that we use here to demonstrate the advantages of LES[64], the input density is fixed to HF, and the learning target is exclusively the energy, allowing us to clearly focus on the effects of LES as a real-space learning strategy for DFAs. At the same time, this does **not** imply that density learning, as a complementary real-space learning strategy, should be abandoned within frameworks where both densities and energies are learning targets. E.g., in DFA development within KS DFT, where both energies and densities are learning targets, loss functions can incorporate the LES term alongside density term (see, e.g., refs. 31,34), or even further sophisticate such loss functions by using the specific link between energy densities and correlation potential[111]. Such combined strategies would leverage strengths of both real-space approaches, which we plan to explore in future work.

While we demonstrate here several advantages of LES over GES, the strength of GES is that it can readily leverage existing global energy data from extensive chemical databases (e.g., GMTKN55). Computing high-level energy densities per particle for every system in such large databases would be impractical and likely unnecessary, particularly given the high transferability potential of LES demonstrated here using data from just eight atoms. Thus, a practical approach would be to design a loss function incorporating both GES and LES terms: employing GES for existing global energy data while strategically complementing it with LES-based training on carefully selected subsets (e.g., a dozen atoms and representative molecules from "Slim" subsets[112] of GMTKN55) to leverage its power for embedding transferability into ML DFAs.

## Methods

### Computational details

All electronic structure calculations were performed using the PySCF 2.3.0 program package[113,114] within the Python coding environment v3.11.4.

For the evaluations in Figs. 1(d), 2(a), and 6 and Supplementary Tables S1, S2, Supplementary Figs. S3, S17–S19, S20 in the SI, we have used *def2-QZVPPD* basis set[115,116], while for the rest, unless specified otherwise, we have used *def2-QZVP* basis set[115]. Our implementation of the MP2 correlation energy density generator uses the Python package for optimizing tensor contractions[88] together with JAX[70] to enable parallelization and high-performance platform agnostic evaluation of the energy densities. For the energy density generation, we also employ the density fitting[86,87] (DF) approximation for MP2 with the *def2-QZVP(PD)-RI* auxiliary basis set. We adapt the same DF code and combine it with the *def2-universal-jkfit* auxiliary basis set to calculate the exchange energy density. In Supplementary Section S3 in the SI, we discuss the effect on the accuracy from the numerical integration with respect to the DFT grid[117] and from the use of DF.

For the regularized correlation energy density data generation, we set $\kappa = 2$ throughout this work unless otherwise specified.

Reference correlation energies are taken from specified references or obtained from CCSD(T) calculations in PySCF.

### Neural Network training

The neural network training was performed with the Pytorch 2.1.2 deep learning library[118]. The input features (Supplementary Section S6 in the SI) were obtained in Python from the HF PySCF output of the given chemical system and pointwise evaluated on the DFT grid[117]. In particular, two FOD features at electronic temperatures $T_1 = 10,000K$ and $T_2 = 25,000K$ were employed for every system. Their implementation is based on the formula from ref. 71, which we divide by the density to obtain a dimensionless quantity. We employed the Adam optimizer[119] in Pytorch for NN training with custom learning rates incorporating a warm-up period and a fixed number of training steps (epochs).

ML2 was trained on eight small closed-shell atoms/ions (H⁻, He, Be, Mg, Ne, Ar, Ca, and Kr). The architecture of the neural network for ML2 has three hidden layers, each consisting of 16 neurons. In ML2, we apply hyperbolic tangent (tanh) activation functions to the NN output layers, bounding the ML2 weights between $-1$ and 1 (for a concrete example illustrating the range of ML2 $w_c(\mathbf{r})$ weights that justifies this choice, see Supplementary Fig. S20 in the SI). For simplicity, the same tanh activation is also used for all hidden layers. All models in this work are optimized by using either the LES-based or the GES-based loss function (see Supplementary Section S8 in the SI for specific details). In ML2, the learning rate follows an exponential decay that continues over 5000 epochs including the warm-up phase.

A detailed overview of the training set for the first MLS2 example is given in Supplementary Section S12 in the SI. For the architecture of the MLS2's NN, we used four hidden layers with 64 neurons each. Following the ML2 architecture, in MLS2 we also use tanh activation function for the hidden layers, while the sigmoid function is applied to the output layer to constrain the MLS2 weights between 0 and 1 before scaling them by a factor of 10 (see Supplementary Section S11 in the SI for further details on MLS2 construction). The total loss function contains total and interaction correlation energy errors, as detailed in

Supplementary Section S8 in the SI. Here, the learning rate follows also an exponential decay over 1000 epochs.

The original MLS2 architecture (4 × 64) is also used for MLS2@W4, and in Supplementary Fig. S18 in the SI, we compare alternative NN architectures and show that 4 × 64 offers modest improvements in validation loss over smaller models, motivating its use. Additionally, we created ten MLS2@W4 NNs with different random seed initializations (see Supplementary Fig. S19 in the SI). From these ten models, the final MLS2@W4 NN shown in Fig. 6 was selected based on the lowest MAE on the entire W4-11 set used for training and validation. Since MLS2@W4 also involves open-shell systems, we have used unrestricted Hartree–Fock (UHF) orbitals. Following ref. 120 of Sim and co-workers, constrained-UHF (CUHF) was employed for reactions where UHF orbitals exhibited spin contamination following the criterion from ref. 121. In Fig. 6, translucent squares behind the markers denote seen energy types for a given model. The datasets shown correspond to the following energy types: atomization energies (W4-11), reaction energies (W4-11RE, BH76RC), noncovalent interaction energies (RG18), barrier heights (BH76), and the separate self-interaction error set (SIE4 × 4). While revDSD-PBEP86-D4 was trained on GMTKN55 (which includes W4-11, SIE4 × 4, BH76, BH76RC and RG18), $\omega$B97M(2) was trained/validated on subsets of Head-Gordon's large database, covering the same energy types with comparable datasets[108] that are marked as seen for this model. For further MLS2@W4 training and testing details, see Supplementary Section S13 in the SI.

## Data availability
Data supporting the key findings of this study are available from the manuscript and its supporting information. Additionally, a Source Data file is provided with this paper. Source data are provided with this paper.

## Code availability
The Python code for the energy density generators used in this work is publicly available at Zenodo[122] under the GNU General Public License v3.0.

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

## Acknowledgements

We thank for fruitful discussions and helpful comments from Kimberly Daas and Tim Gould. We also thank Prof. Eunji Sim for giving us access to their code[120], which we have used to calculate CUHF orbitals in PySCF. The authors acknowledge technical support for the energy density generator from the EPICURE Support team (Emmanuel Kieffer, Frederic Pinel, Alban Rousset and Francesco Bongiovanni) within the EuroHPC project (EHPC-DEV-2024D97-076). The authors also acknowledge funding from the SNSF Starting Grant project to S.V. (TMSGI2_211246).

## Author contributions

E.P. ran all ML simulations. The electronic structure dataset was generated by H.Z. S.V. and E.P. did the data analysis and co-wrote the paper. S.V. supervised the project and conceptualized the idea together with E.P.

## Competing interests

The authors declare no competing interests.
