## [Transparent Peer Review file · Nature Communications]

Real-space machine learning of correlation density functionals

Corresponding Author: Professor Stefan Vuckovic

Version 0:

Reviewer comments:

Reviewer #1

(Remarks to the Author)

In this work by Polak, Zhao, and Vuckovic, the authors present a machine-learning strategy for density functional development where local energy density is used as a fundamental training target. This work is based on their earlier discovery in ref. 67, where the authors produced the so-called MP2 correlation energy density.

In my view, in modern days of density functional development, one may argue that high-impact density functional development should either (1) use minimum empiricism to yield surprisingly high accuracy (e.g., Perdew's SCAN) or (2) rely on increased empiricism to maximize accuracy (e.g., Head-Gordon's wB97 functionals).

I see this work by Vuckovic as belonging to the second category, where one hopes to achieve accuracy beyond existing functionals at the expense of potentially increased empiricism. While I enjoyed reading this work, I think that the method is in its early stage to make an impact on the density functional development. Namely, I would like to see if the authors could produce a density functional based on their scheme that is competitive in accuracy with some of the best density functionals that exist now, such as wB97M(2) and DSD family.

With these in mind, I have the following comments:

(1) I would like to see the authors benchmarking their MLS2 functional against state-of-the-art density functionals over various test sets (GMTKN55 or MGCDB84).

(2) I found it unsatisfactory that the authors chose to use spin-restricted orbitals when investigating bond-breaking problems. I don't know why they couldn't learn same-spin and opposite-spin correlation energy density to handle open-shell and bond-breaking problems. This seems important to address since bond-breaking was investigated.

(3) I noticed that the authors used a regularization parameter of 2. I remember Head-Gordon's original paper proposing 1.45 and then 1.1 in the follow-up paper. I would worry about comparing absolute energies with stronger regularization, but I would guess that relative energy performance would improve (e.g., Fig 5(a)).

(Remarks on code availability)

Reviewer #2

(Remarks to the Author)

Recently, the density functional theory, a basis for the modern first-principles calculation for the materials science, has much benefited from the machine learning technology. Mainly it helped the researchers to explore model exchange-correlation functional forms with enhanced flexibility. The important goal with the ML DFT development is that the training is profitable—the cost for application has to be lower than that for the training. For this purpose, the authors focus on a challenge of transferability, or applicability of the trained functional to unseen materials. The authors hence propose a method to train the local energy density as target. Training with local data has been interested before as it provides massive training data even in a few training materials, as properly acknowledged in the text. The predecessors targeted the charge density distribution whereas the authors tested their new scheme referring to the local energy density.

The authors examined if the correlation part of the energy density, which is of pivotal importance to proceed beyond the hybrid functionals, is well trained in a transferable manner. They formulated the local correlation energy density within a perturbation theory MP2. The MP2 theory is in principle advantageous for incorporating the dispersion force. Although the

local energy density is arbitrary to some extent as the authors are aware, the authors' "gauge" yields successful reproduction of the dissociation curves of the MP2 with the Hartree-Fock computational cost. Also, the authors demonstrate an improvement over the MP2 accuracy toward CCSD(T) utilizing the MP2 reference with the MP2 cost, by introducing the spin-dependent neural-network weight functions. These results are interesting, but there arise some concerns on the manuscript, that makes the referee reluctant to recommendation for publication in Nature Communications.

1)
The referee finds that authors are taking two controversial actions, perhaps without being conscious of their potential controversy. (i) they are abandoning to target the charge density as the physical quantity to be reproduced. Specifically they calculate the HF charge density (and orbitals) and use them as input for the machine-learning surrogate for the high-level approximation (MP2). In the previous "real-space learning" works people learned the charge density, but with the one-shot scheme like the present authors' the charge density CANNOT be taken as the target to learn. This necessity why the authors had to adopt the energy density does not seem properly motivated in the introduction ("... all these works still adhere to the 1 system=1 energy data point approach."). (ii) Another thing is that they argue the correlation energy density relating it with bonding characters like in Fig.1e. Since this quantity is gauge dependent the referee does not think that the sound interpretation is possible.

Especially for the second, the referee is afraid that the publication of the current manuscript in the popular journal like NatCommun may mislead the community. As further examinations accumulate, people may later accept this comparison useful, but currently such studies are lacking. I recommend some more specific journal for the current manuscript, unless the authors can relate the local energy in the present gauge to physical observables in a more reasonable manner. For the latter goal an extensive discussion on authors' implication behind note "transparently arising from the many-body wavefunctions^{62, 64, 67, 68} (see Refs. 67 and 68 for the derivations)" should be essential. For the first point, the authors' action seems to imply an important issue in the field: Whether we SHOULD abandon accuracy for the charge density beyond HF? The authors adopt the density corrected DFT strategy, which is for avoiding "uncontrolled error cancellation". However, due to this the current method cannot go more accurate than the error bound originating from the deviation of the HF density from the ideal density. Such fundamental limitation hidden in the present scheme and why the authors still expose the current method as important should be clearly stated in the text. Besides, the fact has to be noted also that the current scheme's computational cost cannot be lower than the HF cost, inferior to the DFT cost, though it does not decrease the value of the manuscript.

If the referee is not satisfactory with the perhaps newly added materials, he/she will not agree to the acceptance but still believe that the manuscript warrants publication in any other journal, as it presents some interesting elements. The authors adopt the temperature dependent fractional charge density as a descriptor, which appears appropriate for the dispersion force. Leveraging the rough approximation for more accurate correlation by separate introduction of the spin-orientation dependent weight factors is inspiring for further developments. Transferability from isolated atom to small molecules is intriguing.

2)
The comparison between the GES and LES methods seems unfair. The training data are too few (ill-conditioned) for the GES and therefore it is obviously hopeless that the optimized NN parameters reproduces the local values. What the authors have to prove is how much the LES is advantageous, even at the price of the gauge dependence in the quantity to learn. To fairly show this, a result for the GES, which probably yields accurate dissociation curves with enough increased amount of training data, has to be provided. Readers can then judge which is useful; reduced training systems with gauge arbitrariness or enormous training systems without gauge arbitrariness.

Below, the referee points out more specific comments, roughly in order from important to trivial.

3)
Development of energy as well-defined local quantity also has some history, which should be appreciated. For example, refer to Tachibana, J. Mol. Struct.: THEOCHEM 943, 138 (2010).

4)
"We observe in Fig.2a that the σ_s component predominantly contributes to binding (negative regions)," Seems inverted.

5)
Is it possible to improve the authors' energy density formalism to more sophisticated approximations? I think it is possible as long as the total energy is formally written, with e.g. perturbation theory. How about the use with variational methods like diffusion Monte Carlo, in which the local decomposition of the total energy $\langle H \rangle$ is not formally plausible?

(Remarks on code availability)

Reviewer #3

(Remarks to the Author)

In their paper "Real-space machine learning of correlation density functionals", the authors devise a strategy for developing correlation functionals for Kohn-Sham density functional theory (DFT) in terms of machine learning the correlation energy density, i.e. the contributions to the correlation energy, point by point in real space.

This is an interesting approach that I have been waiting to see for some time.

Their work is interesting and a valuable addition to the existing literature. In my opinion, the paper has the potential to be published in Nature Communications, but there are some open questions that should be addressed before making a decision.

The following points should be considered before proceeding further. I have included my comments and questions below, which can also be found in the annotated PDF.

C.1

A few more paper on extended the scales of DFT based on ML should be cited here to better reflect the existing body of literature on this subject:

K. Mills, K. Ryczko, I. Luchak, A. Domurad, C. Beeler, and I. Tamblyn, Chem. Sci. 10, 4129 (2019).

[<https://doi.org/10.1039/C8SC04578J>]

L. Fiedler, N. A. Modine, S. Schmerler, D. J. Vogel, G. A. Popoola, A. P. Thompson, S. Rajamanickam, A. Cangi, Npj Comput. Mater. 9, 115 (2023). [<https://doi.org/10.1038/s41524-023-01070-z2>]

J. A. Rackers, L. Tecot, M. Geiger, and T. E. Smidt, Mach. Learn.: Sci. Technol. 4, 015027 (2023).

[<https://doi.org/10.1088/2632-2153/acb314>]

A. Chandrasekaran, D. Kamal, R. Batra, C. Kim, L. Chen, and R. Ramprasad, npj Comput. Mater. 5, 22 (2019).

[<https://doi.org/10.1038/s41524-019-0162-7>]

C.2

Define the inputs to the neural network s, q, α, f_1, f_2 . It is unclear at this point what these mean.

C.3

While the loss can be measured in terms of absolute values, in practice, it is often defined in terms of a mean squared loss. How are the loss functions for GES and LES implemented?

C.4

In their analysis of MLS2, the authors only partially explain why the MLS2 model performs better than ML2 and κ MP2 mode. While it is clear that the model should perform better, it seems unclear whether this is due to the choice of model in terms of Eq. (11) or whether it is due to the training data used, as the MLS2 model is trained with higher quality data than the ML2 model. However, the authors could check how the higher quality dataset affects the model accuracy by training the ML2 model on the extended dataset as used for MLS2 and then comparing this new model with the results in Fig. 5.

C.5

I have a slightly more detailed but crucial question about the uncertainty and stability of the ML2 and MLS2 models: The authors show predictions of binding curves of their ML models, e.g. in Figures 1c, 1e, 3 and 5. However, they show only one curve for each prediction. Is it a single prediction or an average of predictions? Furthermore, there is an inherent uncertainty associated with neural network predictions. Depending on the random initialization of the neural network before training, the weights will converge to certain values after training. I would therefore like the authors to quantify the uncertainty and stability of their predictions. They could do this by training an ensemble of models that all start with different random initializations. Once trained, each model will produce different predictions. The spread of these predictions would show whether their ML model gives stable predictions. This would be important to see, otherwise the authors might just have been "lucky" with their predictions.

C.6

Furthermore, how did the authors decide on the particular choice of hyperparameters, in particular the two different network architectures and the choice of activation functions in their two ML models?

(Remarks on code availability)

Version 1:

Reviewer comments:

Reviewer #1

(Remarks to the Author)

I was pretty impressed by the work put into this revised manuscript by the authors. I want to take a moment to thank them for addressing my comments. These days, I don't see many authors taking the referee comments seriously and trying their best to improve their manuscript, addressing them.

I am pretty conflicted now in that I appreciate the authors' effort in revising the manuscript, but I didn't find the results

impressive.

The new benchmark study (Fig. 6) presents the performance of MLS2@W4 worse than that of wB97M(2) in all sets but SIE4x4. There are only four distinct chemical species in SIE4x4 with four dimer distances in each species (only 16 data points!). I don't think this qualifies as a success of this approach over a broad range of chemical systems.

One way to assess the impact of the current work is the following: Will the practitioners of DFT be immediately adopting (or even considering the adoption of) this method instead of using state-of-the-art double hybrid functionals? The answer seems to be no, given the current results. Besides, I worry about the cost of this functional since the authors mention that they can't run all the systems in the benchmark sets easily. All things considered (both accuracy and speed), I recommend against publication of this work in Nature Communications.

Minor comments:

(1) I found their wB97M(2) result of H₂⁺ PES in Fig. S17 irreproducible. I get -0.59791473 from wB97M(2) and -0.6019936319 from HF. This should have about 2.56 kcal/mol error, but their Fig. S17 presents ~5 kcal/mol error? I suspect they copied the Q-Chem output energy wrong for wB97M(2). If I take the SCF energy, then I get -0.5937473528, which gives me about 5 kcal/mol error in the end. But this is not the right wB97M(2) energy to take! I recommend the authors double-check their wB97M(2) results. I used "xc_grid 00099000590" for the grid.

(2) The authors seem to present the best trained NN model results for Fig. 6. I appreciate that the authors are open about the training dependence of the model as presented in Fig. S19. However, I would appreciate it if the authors presented the "worst-performing" case as well, along with the best one.

(Remarks on code availability)

Reviewer #2

(Remarks to the Author)

The authors responded to my original comments sincerely. In particular on the gauging problem, the authors made rigorous the relation between the correlation energy density and MP2 theory in newly added Ref.67, thanks to which the meaning of the authors' gauge has been better clarified. Although this does not completely resolve the gauge arbitrariness issue, the authors showed practical usefulness of their gauge by revised Fig.2 ($\Delta \bar{e}_c$ more directly shows where the electron contributes to the stabilization) and Fig.4a (some other gauge choices yield worse transferability). The revised manuscript is almost acceptable but please consider the following comment that is continual to one issue I raised.

1) "For simplicity and to create a challenging transferability test, we train our ML2-based NN only on eight small closed-shell atoms/ions (H-, He, Be, Ne, Mg, Ar, Ca, and Kr)."

I iterate that in the comparisons the GES is hopeless since for the GES model the situation is in very overparametrized condition because of too few training data (Authors used the same model architecture with (# of features x 16)+256+256+16 parameters, right?). Meanwhile, the authors' addition Fig. S13 is informative since the improvement of GES is slow if we increase data to alleviate the overparametrization.

Note that I'm not doubting that LES is useful but was just saying that the comparison was redundant. But I'm getting inclined to an idea that it may be informative for general readership. I would therefore be satisfied if the fact—in the comparisons GES is suffering from overparametrization (or underdetermined condition, in other words)—is written in a well visible place explicitly.

Provided that this point is responded, I will finally recommend Acceptance.

2)

I find "... and it what follows" to be fixed.

(Remarks on code availability)

Reviewer #3

(Remarks to the Author)

The authors have addressed all points raised in my previous report and have particularly demonstrated the robustness of their results. I do not have any further questions or comments and recommend publication of their manuscript.

(Remarks on code availability)

Version 2:

Reviewer comments:

Reviewer #2

(Remarks to the Author)

The revision is satisfactory. I recommend the publication.

(Remarks on code availability)

All changes are detailed below in the point-by-point response, where we also list specific manuscript revisions. Additionally, we provide a marked-up review version of the manuscript, with all changes indicated in blue and with each edit explicitly labeled according to the reviewer's comment number (e.g., "@1.4" corresponds to point 4 raised by referee 1). Furthermore, references to manuscript parts with additions are marked in green below.

1. Points raised by Reviewer 1

1.1 Point raised by Reviewer: In this work by Polak, Zhao, and Vuckovic, the authors present a machine-learning strategy for density functional development where local energy density is used as a fundamental training target. This work is based on their earlier discovery in ref. 67, where the authors produced the so-called MP2 correlation energy density. In my view, in modern days of density functional development, one may argue that high-impact density functional development should either (1) use minimum empiricism to yield surprisingly high accuracy (e.g., Perdew's SCAN) or (2) rely on increased empiricism to maximize accuracy (e.g., Head-Gordon's WB97 functionals). I see this work by Vuckovic as belonging to the second category, where one hopes to achieve accuracy beyond existing functionals at the expense of potentially increased empiricism.

Our answer and changes to the manuscript: We fully agree with the reviewer that our work indeed falls into the category of empirical DFAs, while we stress that our primary focus is on our real space ML strategies targeting the transferability problem of ML-based DFAs (absent in e.g., physically constrained SCAN, mentioned by the reviewer). This limited transferability indeed arises from the increased empiricism of ML DFAs, employed to achieve higher accuracy close to the training domain.

Despite the focus on empirical DFAs, guided by new results (primarily from the Section *Uniqueness and Robustness of our LES-based ML2 model in Results*), we now discuss in more nuanced terms the previously overlooked synergy between empiricism and physical constraints in our models. Specifically, we show that the full advantage of our local energy loss (LES) approach over commonly used global energy loss (GES) can **only** be realized when combined with suitable physical constraints/considerations. These include (i) the physically informed functional form of ML2 in which LES is used, (ii) training explicitly on the correlation energy density per particle (e_c) rather than alternative local energy quantities (such as its density-weighted counterparts), (iii) and the careful choice of e_c gauge. Please see changes in *Introduction* (p. 2) and *Discussion* (p. 11) following the @1.1 label in the marked-up manuscript.

1.2 Points raised by Reviewer: While I enjoyed reading this work, I think that the method is in its early stage to make an impact on the density functional development. Namely, I would like to see if the authors could produce a density functional based on their scheme that is competitive in accuracy with some of the best density functionals that exist now, such as wB97M(2) and DSD family.

With these in mind, I have the following comments:

I would like to see the authors benchmarking their MLS2 functional against state-of-the-art density functionals over various test sets (GMTKN55 or MGCD84).

Our answer and changes to the manuscript: The reviewer raises an important point for our manuscript and for MLS2 developments. While our paper primarily focuses on proposing and analyzing different real-space ML DFA strategies in terms of their transferability (rather than delivering a single final ML DFA), we agree that it is essential to test whether the MLS2 approach can, in principle, compete with state-of-the-art double hybrids as the reviewer suggests.

We remark that the omission of benchmarking involving open-shell systems was due to technical limitations. Specifically, our previous implementation of $e_{c,os}^{kMP2}(\mathbf{r})$ and $e_{c,ss}^{kMP2}(\mathbf{r})$ was restricted to closed-shell systems. Additionally, applying MLS2 to the entire GMTKN55 database, particularly for larger molecules, would require optimization of our code that would by far exceed several months. This challenge, common in DFA developments relying on new ingredients whose infrastructure is still evolving (such as our energy density generators), is very different from DFA developments using mature infrastructure as some of us have recently discussed in a separate publication (Ref. 110 = doi.org/10.1021/acs.jctc.5c00512). This is why we could not yet apply our methodology to the entire GMTKN55 database. **Nevertheless, during the revision period, we have devoted substantial additional implementation effort to extend our code to open-shell systems. This has enabled us to benchmark MLS2 against representative GMTKN55 subsets, sufficient to answer how competitive the MLS2 strategy is compared to state-of-the-art double hybrids.**

To address the reviewer's point while preserving our manuscript's nature of exploring real-space ML DFA strategies in the transferability context, we have trained MLS2 on the W4-11 atomization energies by adding a spin-polarization feature to it (MLS2@W4) and tested the resulting model's transferability on **unseen** representative GMTKN55 subsets, namely: open-shell cationic dimers (SIE4x4), barrier heights (BH76), non-covalent interactions (RG18), reaction energies (BH76RC), and the W4-11-RE dataset, which contains ~11k

reaction energies derived from W4-11 total energies. For these sets, we compared the performance of MLS2@W4 against the two double hybrids proposed by the reviewer, again **in the context of relative transferability of underlying methods (their training versus used tests)** we show that MLS2@W4 is competitive with these double hybrids and greatly improves upon them for the challenging SIE4x4 set containing self-interaction problems. This comment of the reviewer also prompted us to realize that unlike standard double hybrids, our MLS2 construction is exact for one-electron systems. For further details, see newly added material in Sec. Results/Spin-resolved and regularized modelling of the correlation energy density (p. 10 and p. 11) including Fig. 6, following the @1.2 label in the marked-up manuscript.

Other minor changes to the manuscript: In *Methods* (p. 13) and Sec. S13 in SI, we provide additional details regarding the construction of MLS2@W4.

1.3 Point raised by Reviewer: I found it unsatisfactory that the authors chose to use spin-restricted orbitals when investigating bond-breaking problems. I don't know why they couldn't learn same-spin and opposite-spin correlation energy density to handle open-shell and bond-breaking problems. This seems important to address since bond-breaking was investigated.

Our answer and changes to the manuscript: The reviewer raises an important concern here, and we would like to make two distinct points in response:

1. In relation to their previously raised point 1.2, we have extended our energy densities to open-shell systems and built a spin-polarized MLS2@W4, thus going beyond spin-restricted orbitals.

2. However, for bond-breaking of closed-shell molecules for which we show total energy dissociation curves (e.g., new Fig. 4 and Fig 5), we intentionally used spin-restricted orbitals in the context of the challenge of correctly describing bond-breaking without lowering energies via spin-symmetry breaking (Refs. 94-96 contextualize this challenge). We failed to communicate this in the previous manuscript version, and we now contextualize this clearly (see the text and references additions in newly added Sec. *Result/Uniqueness and Robustness of our LES-based ML2 model* in on p. 8, following the @1.3 label in the marked-up manuscript).

1.4 Point raised by Reviewer: I noticed that the authors used a regularization parameter of 2. I remember Head-Gordon's original paper proposing 1.45 and then 1.1 in the follow-up paper. I would worry about comparing absolute energies with stronger regularization, but I would guess that relative energy performance would improve (e.g., Fig 5(a)).

Our answer: This is indeed an important remark, as our previous manuscript did not justify our choice of $\kappa = 2.0$ for generating $e_c^{\kappa\text{MP2}}(\mathbf{r})$ as the proxy reference for training our LES-based ML2 model. Dissociation curves of diatomics are a key test of this model's transferability, and κMP2 removes the energetic divergences of MP2 at large bond lengths in these curves. However, we find that using $\kappa = 2.0$ in these curves significantly improves the total energies at stretched bond lengths compared to the originally proposed $\kappa = 1.4$ based on energy differences [Ref. 71]. At the same time, $\kappa = 2.0$ and $\kappa = 1.4$ yield comparable accuracy near equilibrium. To illustrate this tradeoff, we have added Fig. S1 in the SI for the N_2 dissociation curve. In the revised manuscript, we now reference this figure and explain our choice of $\kappa = 2.0$ (see Sec. *Results / Improving and deriving PT2 correlation energy densities per particle* on p. 4 following the @1.4 label in the marked-up manuscript).

2. Points raised by Reviewer 2

2.1 Point raised by Reviewer: Recently, the density functional theory, a basis for the modern first-principles calculation for the materials science, has much benefited from the machine learning technology. Mainly it helped the researchers to explore model exchange-correlation functional forms with enhanced flexibility. The important goal with the ML DFT development is that the training is profitable—the cost for application has to be lower than that for the training. For this purpose, the authors focus on a challenge of transferability, or applicability of the trained functional to unseen materials. The authors hence propose a method to train the local energy density as target. Training with local data has been interested before as it provides massive training data even in a few training materials, as properly acknowledged in the text. The predecessors targeted the charge density distribution whereas the authors tested their new scheme referring to the local energy density.

The authors examined if the correlation part of the energy density, which is of pivotal importance to proceed beyond the hybrid functionals, is well trained in a transferable manner. They formulated the local correlation

energy density within a perturbation theory MP2. The MP2 theory is in principle advantageous for incorporating the dispersion force. Although the local energy density is arbitrary to some extent as the authors are aware, the authors' "gauge" yields successful reproduction of the dissociation curves of the MP2 with the Hartree-Fock computational cost. Also, the authors demonstrate an improvement over the MP2 accuracy toward CCSD(T) utilizing the MP2 reference with the MP2 cost, by introducing the spin-dependent neural-network weight functions. These results are interesting, but there arise some concerns on the manuscript, that makes the referee reluctant to recommendation for publication in Nature Communications.

Our answer: We thank Reviewer 2 for recognizing the importance of our work, and we greatly appreciate their insightful criticism, which has enabled us to add new material that sheds light on key concepts underlying the developed real-space ML of DFAs. After incorporating these changes, we firmly believe our manuscript is now suitable for publication in Nature Communications.

2.2 Point raised by Reviewer: The referee finds that authors are taking two controversial actions, perhaps without being conscious of their potential controversy. (i) they are abandoning to target the charge density as the physical quantity to be reproduced. Specifically, they calculate the HF charge density (and orbitals) and use them as input for the machine-learning surrogate for the high-level approximation (MP2). In the previous "real-space learning" works people learned the charge density, but with the one-shot scheme like the present authors' the charge density CANNOT be taken as the target to learn. This necessity why the authors had to adopt the energy density does not seem properly motivated in the introduction ("... all these works still adhere to the 1 system=1 energy data point approach.")

Our answer: The reviewer is correct on this point, which was unclear in the previous manuscript version, causing confusion about our intent regarding LES vs density learning as two distinct real-space ML strategies. This confusion partly arose from a lack of clarity regarding the theoretical framework employed to demonstrate LES. We explicitly clarify this framework in parts A below, while part B addresses directly the relationship between LES and density learning.

A. While the LES strategy itself is general, we now clarify that our implementation here specifically targets DFAs for correlation energy defined relative to the HF reference (exact minus HF energy). Following the reviewer's criticism, we have published a separate paper (newly added Ref. 67 = doi.org/10.1021/acs.jctc.5c00348), which establishes a formal framework for correlation energy densities based on the Møller–Plesset adiabatic connection (MPAC), that integrate to the targeted correlation energy. Within this framework, the MP2 energy densities used in the present work emerge naturally as the weakly interacting limit and share the same gauge as their exact counterparts. With this foundation established, in the revised manuscript we emphasize that our current approach specifically targets correlation DFAs, and the use of HF densities as inputs for such functionals is formally grounded in MPAC theory. Please see *Sec. Results / Improving and deriving PT2 correlation energy densities per particle* on p. 3 following the @2.2.A label in the marked-up manuscript as well as newly added *Sec. S1* in the SI that gives detailed derivation. The summary of this has been also put in the Introduction accordingly (see newly added text in Introduction on p. 3 following the @2.2.A label in the marked-up manuscript). This foundation clarifies our manuscript and avoids conflating our framework with "density-corrected DFT," as noted by the reviewer in point 2.4 below, which we now explicitly clarify above Eq. 4 on p. 3 and p. 4, following the @2.2.A label in the marked-up manuscript.

B. Our correlation DFA construction is based on the use of HF density, with the theoretical justification for this choice grounded in MPAC theory. Thus, the reviewer is fully right that the underlying scheme is non-self-consistent ("one-shot") where the density is not a learning target. **While this scheme enables us to isolate the effects of LES by using energy as the sole learning target, we now explicitly clarify that we do not advocate abandoning density learning in the frameworks where this is applicable. Indeed, we stress that in frameworks targeting both densities and energies (e.g., KS DFT-based DFAs, as opposed to the MPAC-based DFAs here), LES is intended to complement rather than replace density learning to enhance DFA transferability.** This point is clarified at several places in the manuscript, please see: (i) p 3, *Introduction*, the sentence following the @2.2.B label in the marked-up manuscript; (ii) p 12, *Discussion*; the whole paragraph following the @2.2.B label in the marked-up manuscript.

2.3 Point raised by Reviewer: (ii) Another thing is that they argue the correlation energy density relating it with bonding characters like in Fig.1e. Since this quantity is gauge dependent the referee does not think that the sound interpretation is possible.”

Our answer: The reviewer is right, as the previous manuscript version missed the link between interaction energy densities and the underlying physics. We note that our plots of interaction energy densities are used for weak (noncovalent) interactions with particular emphasis on dispersion, not bonds (e.g., covalent) in general. **We now add material clarifying that despite gauge dependence, our chosen gauge is physically meaningful for weak interactions, as detailed in part C below.** Parts A and B introduce other changes we had to make to set the foundation for part C.

A. Although this point is separate from the gauge issue addressed below (part C below), following the reviewer’s comment, we realized that a meaningful interaction plots require plotting interaction correlation energy densities [$\Delta\bar{e}_c(\mathbf{r})$], rather than their per-particle counterparts [$\Delta e_c(\mathbf{r})$], as the latter lacks a well-defined density factor to integrate properly to ΔE_c (the correlation component of the interaction energy). To implement this change, we clearly distinguished $e_c(\mathbf{r})$ and $\bar{e}_c(\mathbf{r})$, early in the manuscript, introduced their interaction (Δ) counterparts, and used $\Delta\bar{e}_c(\mathbf{r})$ for visualization whenever interaction quantities appear (Fig.1(d) bottom and Fig. 2). These changes are made below Eqs. 1, 2, 3 (p. 3) and below Eq. 4 following the @2.3.A label in the marked-up manuscript.

B. To set the stage for part C below and link this response to the theoretical framework resulting from the response to point, we now define $e_c(\mathbf{r})$ and $\Delta\bar{e}_c(\mathbf{r})$ (specifically their κ -generalized counterparts) in terms of correlation contributions from the underlying pair densities (see newly added material around Eqs. 5 and 10 on p. 4, following @2.3.B label in the marked-up manuscript and Sec. S1 in the SI).

C. With the stage set in A and B, we now finally address the reviewer’s point by introducing new section in *Results/Regularized PT2 correlation energy densities for interaction energies on p. 5*. **This whole section is devoted to this point (@2.3.C)**. There, we illustrate the physics of $\Delta\bar{e}_c^{MP2}(\mathbf{r})$ through its link to the *interaction correlation pair density* by using the Helium dimer as a simple yet key dispersion-bound example. Building upon this analysis, in the same section we clarify that **even though $\Delta\bar{e}_c(\mathbf{r})$ is not uniquely defined, the specific gauge we use yields a physically interpretable local correlation energy quantity for describing weak interactions between fragments**. The newly added Fig. S3 in the SI also supports *Sec. Results/Regularized PT2 correlation energy densities for interaction energies*.

As a brief note also relevant to point 2.3.A above, we also highlight in the second sentence of *Sec. Results/Regularized PT2 correlation energy densities for interaction energies on p. 5* a subtle yet crucial distinction between the use of $e_c(\mathbf{r})$ vs. $\bar{e}_c(\mathbf{r})$.

2.4 Point raised by Reviewer: Especially for the second, the referee is afraid that the publication of the current manuscript in the popular journal like NatCommun may mislead the community. As further examinations accumulate, people may later accept this comparison useful, but currently such studies are lacking. I recommend some more specific journal for the current manuscript, unless the authors can relate the local energy in the present gauge to physical observables in a more reasonable manner. For the latter goal an extensive discussion on authors’ implication behind note “transparently arising from the many-body wavefunctions^{62, 64, 67, 68} (see Refs. 67 and 68 for the derivations)” should be essential. For the first point, the authors’ action seems to imply an important issue in the field: Whether we SHOULD abandon accuracy for the charge density beyond HF? The authors adopt the density corrected DFT strategy, which is for avoiding “uncontrolled error cancellation”. However, due to this the current method cannot go more accurate than the error bound originating from the deviation of the HF density from the ideal density. Such fundamental limitation hidden in the present scheme and why the authors still expose the current method as important should be clearly stated in the text. Besides, the fact has to be noted also that the current scheme’s computational cost cannot be lower than the HF cost, inferior to the DFT cost, though it does not decrease the value of the manuscript.

If the referee is not satisfactory with the perhaps newly added materials, he/she will not agree to the acceptance but still believe that the manuscript warrants publication in any other journal, as it presents some interesting elements. The authors adopt the temperature dependent fractional charge density as a descriptor, which appears

appropriate for the dispersion force. Leveraging the rough approximation for more accurate correlation by separate introduction of the spin-orientation dependent weight factors is inspiring for further developments. Transferability from isolated atom to small molecules is intriguing.

Our answer and changes to the manuscript: We thank to the reviewer for identifying contributions of our work and critical comments that enables us to greatly improve our work. The responses to specific comments are given below.

A. In relation to:

“unless the authors can relate the local energy in the present gauge to physical observables in a more reasonable manner. For the latter goal an extensive discussion on authors’ implication behind note “transparently arising from the many-body wavefunctions^{62, 64, 67, 68} (see Refs. 67 and 68 for the derivations)” should be essential”,

we have addressed this point in detail in **2.3** response above, which concerns the link of our gauge to dispersion interactions. We note that the connection to wavefunction-based quantity (pair-density) has been indeed “essential” in this response. Second, we also believe that the learnability of our gauge vs other gauges had been very important step to strengthen our construction (see part **D** below).

B. In relation to:

“For the first point, the authors’ action seems to imply an important issue in the field: Whether we SHOULD abandon accuracy for the charge density beyond HF? The authors adopt the density corrected DFT strategy, which is for avoiding “uncontrolled error cancellation”.

However, due to this the current method cannot go more accurate than the error bound originating from the deviation of the HF density from the ideal density. Such fundamental limitation hidden in the present scheme and why the authors still expose the current method as important should be clearly stated in the text.”,

we have addressed this point in detail in **2.2** response above.

C. In relation to:

“Besides, the fact has to be noted also that the current scheme’s computational cost cannot be lower than the HF cost, inferior to the DFT cost, though it does not decrease the value of the manuscript.”,

we have addressed this comment on p. 6 below Eq. 13 (see the text addition following the @2.4 label in the marked-up manuscript).

D. Regarding gauge dependence and LES, we added further material concerning **learnability of our gauge** motivated by the reviewer’s remark: *“Transferability from isolated atom to small molecules is intriguing.”* Specifically, we tested whether this transferability from atoms to small molecules is preserved by LES with other gauges, which we obtained through a controllable transformation (Eq. 14) that preserves integrated correlation energies (see Eq. 14, newly added Fig. 4(a), and the added text in newly added Sec. *Results/Uniqueness and Robustness of our LES-based ML2 model in Results* on p. 8, following the @2.4 label in the marked-up manuscript, as well as additional Figs. S7 and S8 in the SI). Crucially, this transferability holds only with our original gauge and can be lost with gauge changes. **Although Eq. 14 does not cover all possible gauges, this clearly shows that transferability of LES-based ML2 arises not only from LES data efficiency but also critically depends on our chosen gauge.**

2.5. Point raised by Reviewer: The comparison between the GES and LES methods seems unfair. The training data are too few (ill-conditioned) for the GES and therefore it is obviously hopeless that the optimized NN parameters reproduces the local values. What the authors have to prove is how much the LES is advantageous, even at the price of the gauge dependence in the quantity to learn. To fairly show this, a result for the GES, which probably yields accurate dissociation curves with enough increased amount of training data, has to be provided. Readers can then judge which is useful; reduced training systems with gauge arbitrariness or enormous training systems without gauge arbitrariness.

Our answer:

A. Our main point with ML2 transferability was to show that, given the same small training set of atoms, LES-based ML2 transfers to diatomic dissociation curves, while GES-based ML2 does not. Nevertheless, the reviewer makes a valid point here by asking what happens once the training set size is increased.

To address the reviewer's point, we designed the following test: we chose one system's dissociation curve as the "primary test", BeH^+ , which had been used already in Fig. 4(c), and progressively increased the training set, starting from our original 8 atoms, then adding 7 points around equilibrium for each dimer in the list [H_2 , LiH , Li_2 , BH , CH^+ , N_2], and ending with BeH^+ itself. Only in this sense, BeH^+ is the "primary test", as its own datapoints are added last. For each new addition of training data points, we retrained NNs for both LES- and GES-based ML2 and observed what happens with the resulting dissociation curves of BeH^+ and other dimers against the κMP2 reference. **While the final training set and additions at each step may not be large in absolute terms, they are major relative to the test (e.g., for the full BeH^+ dissociation curve test, the final model includes in the training its own BeH^+ points near-equilibrium) and the original atomic training set.**

These results are presented in Fig. S13 in the SI, where the LES-based ML2 trained on atoms already gives highly accurate dissociation curves and shows virtually no change as more training data is added to LES-based ML2. In contrast, GES-based ML2, even after these extra points are added to the training set, is still outperformed by LES. While GES-based curves are improving after more data are added to the training, one can still observe unphysical discontinuities in some of the GES-based curves. Even when the last batch of training data is added to create the most "data rich" model that includes its own equilibrium BeH^+ datapoints, unlike LES-based, GES-based ML2 is still not sufficiently accurate for BeH^+ stretched geometries. Using the same models with increasing training data and applying them to the dissociation curves of three additional dimers shown in Fig. S13(b)-(c), similar trends are observed.

Finally, towards the end of Sec. S10 in the SI (the section where these results are added and discussed), we note that although the GES-based ML2 employs the global energy loss (Eq. 2) for training, it still uses the ML2 energy density per particle ansatz (Eq. 13), constructed to satisfy the scaling constraint dictated by $e_c^{\kappa\text{MP2}}(\mathbf{r})$ within our chosen gauge. Consequently, GES-based ML2 is still partially informed by our energy densities per particle gauge, which in turn improves its transferability. Crucially, if instead of this GES that we use here one employs "raw" GES—i.e., using Eq. 2 to learn energy densities without the ansatz of Eq. 12, the resulting GES transferability would be even worse than shown here.

In addition to this analysis in Sec. S10 and Fig. S13 in the SI, and the main conclusion from this analysis shown in, is also referenced in the main paper, in newly added Sec. *Results/Uniqueness and Robustness of our LES-based ML2 model* in Results on p. 9, following the @2.5 label in the marked-up manuscript.

B. In light of the reviewer's comment and the Part A response, where LES and GES are contrasted, we also felt that it is important to stress their potential complementarity when used together in training (see *Discussion* on p. 12 following the @2.5 label in the marked-up manuscript, which also highlights the challenges of applying LES to large existing databases and possible ways to mitigate them).

2.6 **Point raised by Reviewer:** Below, the referee points out more specific comments, roughly in order from important to trivial. Development of energy as well-defined local quantity also has some history, which should be appreciated. For example, refer to Tachibana, J. Mol. Struct.: THEOCHEM 943, 138 (2010).

Our answer and changes to the manuscript: The reviewer is right. We have used this and related references to better contextualize local energy quantities (see the addition in *Introduction Results* on p. 3, following the @2.6 label in the marked-up manuscript).

2.7 **Point raised by Reviewer:** We observe in Fig.2a that the o_s component predominantly contributes to binding (negative regions),
Seems inverted."

Our answer: This issue has already been addressed through our changes in response to comment 2.3 above.

2.8 **Point raised by Reviewer:** Is it possible to improve the authors' energy density formalism to more sophisticated approximations? I think it is possible as long as the total energy is formally written, with e.g. perturbation theory. How about the use with variational methods like diffusion Monte Carlo, in which the local decomposition of the total energy $\langle H \rangle$ is not formally plausible?

Our answer and changes to the manuscript: The reviewer raises an important point. Following the procedure from our new Ref. 67 (doi.org/10.1021/acs.jctc.5c00348; see response to point 2.2A above, which inspired this addition), one can obtain MPAC-based $e_c(\mathbf{r})$ from variational wavefunctions, explicitly demonstrated by using full configuration interaction (FCI) method. We now highlight this approach in *Discussion* (see p. 11, following

the @2.8 label in the marked-up manuscript) as a critical step toward going beyond the current κ MP2-based training targets when applying LES.

Points raised by Reviewer 3.

3.1. Point raised by Reviewer: In their paper "Real-space machine learning of correlation density functionals", the authors devise a strategy for developing correlation functionals for Kohn-Sham density functional theory (DFT) in terms of machine learning the correlation energy density, i.e. the contributions to the correlation energy, point by point in real space. This is an interesting approach that I have been waiting to see for some time.

Their work is interesting and a valuable addition to the existing literature. In my opinion, the paper has the potential to be published in Nature Communications, but there are some open questions that should be addressed before making a decision.

The following points should be considered before proceeding further. I have included my comments and questions below, which can also be found in the annotated PDF.

Our answer: We thank the reviewer for acknowledging the potential of our work and for their valuable feedback in the points below that allowed us to extend the scope of the manuscript, particularly when it comes to the previously overlooked robustness of our ML2 model.

3.2 Point raised by Reviewer: A few more paper on extended the scales of DFT based on ML should be cited here to better reflect the existing body of literature on this subject:

K. Mills, K. Ryczko, I. Luchak, A. Domurad, C. Beeler, and I. Tamblyn, Chem. Sci. 10, 4129 (2019).

[<https://doi.org/10.1039/C8SC04578J>]

L. Fiedler, N. A. Modine, S. Schmerler, D. J. Vogel, G. A. Popoola, A. P. Thompson, S. Rajamanickam, A. Cangi, Npj Comput. Mater. 9, 115 (2023). [<https://doi.org/10.1038/s41524-023-01070-z2>]

J. A. Rackers, L. Tecot, M. Geiger, and T. E. Smidt, Mach. Learn.: Sci. Technol. 4, 015027 (2023).

[<https://doi.org/10.1088/2632-2153/acb314>]

A. Chandrasekaran, D. Kamal, R. Batra, C. Kim, L. Chen, and R. Ramprasad, npj Comput. Mater. 5, 22 (2019).

[<https://doi.org/10.1038/s41524-019-0162-7>]

Our answer and changes to the manuscript: We thank the reviewer for these references that indeed help to better contextualize our work and we have added these citations in *Introduction* (p. 1, references added following the @3.2. label in the marked-up manuscript).

3.3 Point raised by Reviewer: Define the inputs to the neural network s , q , α , f_1 , f_2 . It is unclear at this point what these mean.

Our answer and changes to the manuscript: The reviewer is right, as the feature definitions have been missing. Thus, we have added a brief explanation of the input features for the neural network of Fig. 1(b) in the caption of the figure (see text therein following the @3.3 label in the marked-up manuscript), and detailed mathematical definitions of all the used input features in Sec. S6 in the SI.

3.4. Point raised by Reviewer: While the loss can be measured in terms of absolute values, in practice, it is often defined in terms of a mean squared loss. How are the loss functions for GES and LES implemented?

Our answer: This is an important point, as mean squared loss is often used in ML practices. In this work, for both LES and GES, we have employed the loss in terms of absolute differences as in Eq. 3, with specific implementation details provided in Sec. S8 in the SI, and following the reviewer's comment we now explicitly reference this section in *Methods* (see the text addition on p. 12, following the @3.4 label in the marked-up manuscript).

Following the reviewer's suggestion, we have additionally tested what happens if, instead of absolute-value LES, we use the mean squared loss defined in Eq. S23 (LES^2). We found that using LES^2 instead of LES within the ML2 model introduces very minor changes to its transferability. We have included these new results in new Fig. S12, and reference their conclusions in the main manuscript in the newly added section *Results/Uniqueness and Robustness of our LES-based ML2 model* (see the text addition in that section on p. 9 following the @3.4 label in the marked-up manuscript).

3.5. Point raised by Reviewer: In their analysis of MLS2, the authors only partially explain why the MLS2 model performs better than ML2 and κ MP2 mode. While it is clear that the model should perform better, it seems unclear whether this is due to the choice of model in terms of Eq. (11) or whether it is due to the training data used, as the MLS2 model is trained with higher quality data than the ML2 model. However, the authors could check how the higher quality dataset affects the model accuracy by training the ML2 model on the extended dataset as used for MLS2 and then comparing this new model with the results in Fig. 5.

Our answer: There are several aspects to this question. ML2 and MLS2 serve distinct purposes in this work: ML2 demonstrates the advantage of LES for embedding transferability in DFAs, whereas MLS2 demonstrates the use of Eq. 15 in ML of DFAs (previously Eq. 11). Due to technical constraints, ML2 is currently trained only on the proxy κ MP2 reference, while MLS2 is trained on higher-quality CCSD(T) data but with GES-based training. Thus, a direct *apples-to-apples* comparison between ML2 and MLS2 cannot be performed, as they have different learning targets.

We also note that training ML2 on higher-quality [CCSD(T)] data, as suggested by the reviewer, is precluded by ML2's current physically constrained form, specifically tailored for (κ)MP2 energy densities per particle satisfying Eq. 12, a property not shared by their exact counterparts. Nevertheless, the reviewer's suggestion could still be pursued by constructing an adjusted ML2-like model (e.g., modifying Eq. 13 and incorporating density-based features), thus making it compatible with higher-quality data. However, this would constitute a distinct new research direction. We now mention this possibility in Discussion on p. 11, along with an alternative strategy, which we find more feasible for the future, where MLS2 is built where instead of the exact κ MP2 components entering Eq. 15, one uses its LES-based ML2 surrogates (see comments on p. 11, following the @3.5 label in the marked-up manuscript).

The second point previously overlooked in the manuscript is the regularizing role of Eq. 15 (previously Eq. 11), mentioned by the reviewer, which we believe is a crucial factor underlying the strong performance of MLS2 and MLS2@W4. This regularizing role of Eq. 15 is now discussed towards the end of Sec. Results/Spin-resolved and regularized modelling of the correlation energy density (p. 11) following the @3.5 label in the marked-up manuscript).

3.6. Point raised by Reviewer: I have a slightly more detailed but crucial question about the uncertainty and stability of the ML2 and MLS2 models: The authors show predictions of binding curves of their ML models, e.g. in Figures 1c, 1e, 3 and 5. However, they show only one curve for each prediction. Is it a single prediction or an average of predictions? Furthermore, there is an inherent uncertainty associated with neural network predictions. Depending on the random initialization of the neural network before training, the weights will converge to certain values after training. I would therefore like the authors to quantify the uncertainty and stability of their predictions. They could do this by training an ensemble of models that all start with different random initializations. Once trained, each model will produce different predictions. The spread of these predictions would show whether their ML model gives stable predictions. This would be important to see, otherwise the authors might just have been "lucky" with their predictions.

Our answer and changes to the manuscript: The reviewer mentions a very important aspect of our models in terms of their robustness and stability that we have failed to communicate precisely in the previous manuscript version. Exactly how the reviewer suggests, we tested our ML2 and MLS2 results with respect to training uncertainties by varying the initial seed. We have then extended the same analysis to different neural network architectures to obtain a more complete picture.

For ML2, in new Fig. 4(c) with further results in Figs. S10 in the SI, we add additional energy curves that come from different random seed initialization. The newly made observations are summarized in a new subsection: Results/ Uniqueness and Robustness of the LES-based ML2 model (see the text additions in this section, p. 9 following the @3.6 label in the marked-up manuscript). These results clearly confirm the stability and robustness of the LES-based ML2 model relative to its GES-based counterpart, ensuring that the successful extrapolations of the former are not a "lucky" coincidence. We thank the reviewer for this suggestion as it has enabled us to extend the scope of the paper by further highlighting the previously overlooked advantage of our ML2 model.

Regarding our newest model, MLS2@W4 (Fig. 6, p. 10), we performed ten runs with different initial seeds, and we have included in Fig. S19 in the SI the analogue of Fig. 6 for all ten runs. In Methods (p. 12 and p. 13) following the @3.6 label in the marked-up manuscript, we specify that the final model was selected from these runs based on the lowest MAE on the W4-11 dataset (the "seen set" for MLS2@W4).

3.7 Point raised by Reviewer: Furthermore, how did the authors decide on the particular choice of hyperparameters, in particular the two different network architectures and the choice of activation functions in their two ML models?

Architecture. For ML2, we chose a small NN architecture to facilitate extensive numerical experiments comparing LES and GES. We started with a 3×16 architecture as in Ref. 55, which already provided the desired training-to-validation loss ratio (Fig. S4 in SI) and was used for the final ML2 model. Moreover, as detailed in our response to comment 3.6 and the corresponding additional material, ML2 is highly robust: essentially identical results and consistent conclusions regarding LES vs. GES hold even when using smaller (e.g., 3×8) or larger (e.g., 5×16) NNs, as demonstrated in Figs. 4(d) and S11 in the SI.

For MLS2, we employed a larger NN architecture (4×64), and for our newest MLS2 model, MLS2@W4, we explored the model's performance across different NN architectures (see newly added Fig. S18 in SI). In Methods (p. 12), following the @3.7 label in the marked-up manuscript, we refer to Fig. S18 and explain how these results motivated the choice of the 4×64 architecture.

Activation functions. Following the reviewer's comment, we also better explain the choice of activation functions in the revised manuscript. For ML2, *tanh* activation was applied to the output layers to constrain the outputs appropriately (see text additions in Methods on p. 12, following the @3.7. label in the marked-up manuscript and newly added Fig. S20 in the SI supporting this text), and the same *tanh* was also applied to the hidden layers of ML2.

Following the ML2 architecture, MLS2 also employs *tanh* activation functions for the hidden layers, but we found it more appropriate to constrain the initial MLS2 weights by the sigmoid function before scaling them; hence, this activation was applied to the output layers of MLS2. This reasoning is now better explained in Methods (p. 12) and Sec. Results/Spin-resolved and regularized modelling of the correlation energy density (p. 10), in the sentences following the @3.7. label in the marked-up manuscript.

Response Letter

Please note that in addition to the revised manuscript, we provide a marked-up review version of the manuscript with all changes indicated in blue and with each edit explicitly labeled according to the reviewer's comment number (e.g., "@1.2" corresponds to point 2 raised by Reviewer 1). Furthermore, references to manuscript parts with additions are marked in green below.

1. Points raised by Reviewer 1

1.1 Point raised by Reviewer: I was pretty impressed by the work put into this revised manuscript by the authors. I want to take a moment to thank them for addressing my comments. These days, I don't see many authors taking the referee comments seriously and trying their best to improve their manuscript, addressing them. I am pretty conflicted now in that I appreciate the authors' effort in revising the manuscript, but I didn't find the results impressive.

The new benchmark study (Fig. 6) presents the performance of MLS2@W4 worse than that of wB97M(2) in all sets but SIE4x4. There are only four distinct chemical species in SIE4x4 with four dimer distances in each species (only 16 data points!). I don't think this qualifies as a success of this approach over a broad range of chemical systems. One way to assess the impact of the current work is the following: Will the practitioners of DFT be immediately adopting (or even considering the adoption of) this method instead of using state-of-the-art double hybrid functionals? The answer seems to be no, given the current results.

Besides, I worry about the cost of this functional since the authors mention that they can't run all the systems in the benchmark sets easily. All things considered (both accuracy and speed), I recommend against publication of this work in Nature Communications.

Our answer: We thank the Reviewer for acknowledging our extensive changes during the last revision.

In relation to the critical comments, we would like to emphasize two points concerning the main goals of the paper and the contextualization of the Fig. 6 results:

(1) Our work introduces a fundamentally new way of performing ML for DFAs through the ML2 and MLS2 frameworks based on real-space learning, aimed at overcoming the transferability problem in ML of DFAs. The DFAs presented here serve as demonstrations of how these frameworks can embed transferability in ML of DFAs.

(2) The transferability of MLS2 shown in Fig. 6 directly supports (1). MLS2@W4, as one example of the MLS2 framework, transfers to *unseen* datasets even if it underperforms by ca. 1–2 kcal/mol compared to double hybrids for which those energy types are *seen*. The key point is the distinction between seen and unseen data: had we trained MLS2 on all sets in Fig. 6, its performance would trivially improve. However, our goal was to test transferability to unseen cases and not to introduce just another double hybrid, but to build a framework that enables ML of transferable DFAs. This is what Fig. 6 presents. The revised version of this figure highlights this point more clearly, along with the minor textual additions in Sec. *Methods / Spin-resolved and regularized modelling of the correlation energy*

density (see the text following the @1.1 label in the marked-up manuscript, p. 11).

Furthermore, a comment regarding the cost of MLS2 has been added to *Discussion* (see the text following the @1.1 label in the marked-up manuscript, p. 11).

1.2 Point raised by Reviewer: Minor comments: (1) I found their wB97M(2) result of H₂⁺ PES in Fig. S17 irreproducible. I get -0.59791473 from wB97M(2) and -0.6019936319 from HF. This should have about 2.56 kcal/mol error, but their Fig. S17 presents ~5 kcal/mol error? I suspect they copied the Q-Chem output energy wrong for wB97M(2). If I take the SCF energy, then I get -0.5937473528, which gives me about 5 kcal/mol error in the end. But this is not the right wB97M(2) energy to take! I recommend the authors double-check their wB97M(2) results. I used "xc_grid 00099000590" for the grid.

Our answer: The Reviewer noticed a discrepancy in Fig. S17, which was due to an error in the axis labeling: the interatomic distances were printed in Ångström but mistakenly labeled in as Bohr. We have corrected this mistake and revised Fig. S17 with the proper units. Attached to this response letter is a zip file with all input and output files from Q-Chem, as well as a table with the corresponding energy results in kcal/mol.

1.3 Point raised by Reviewer: The authors seem to present the best trained NN model results for Fig. 6. I appreciate that the authors are open about the training dependence of the model as presented in Fig. S19. However, I would appreciate it if the authors presented the "worst-performing" case as well, along with the best one.

Our answer: While Fig. S19 already illustrates the results from multiple runs with different random seed initializations, we do not find it suitable to present the discarded runs as stand-alone results in Fig. 6. As described in Methods, the final MLS2@W4 model shown in Fig. 6 was selected as the run with the lowest MAE on the full W4-11 set used for training and validation, ensuring that all other datasets in Fig. 6 remain unseen for the final model.

Points raised by Reviewer 2

2.1 Point raised by Reviewer: The authors responded to my original comments sincerely. In particular on the gauging problem, the authors made rigorous the relation between the correlation energy density and MP2 theory in newly added Ref.67, thanks to which the meaning of the authors' gauge has been better clarified. Although this does not completely resolve the gauge arbitrariness issue, the authors showed practical usefulness of their gauge by revised Fig.2 ($\Delta \bar{e}_c$ more directly shows where the electron contributes to the stabilization) and Fig.4a (some other gauge choices yield worse transferability). The revised manuscript is almost acceptable but please consider the following comment that is continual to one issue I raised

Our answer: We are thankful for the Reviewer's recognition of our efforts.

2.2 Point raised by Reviewer: 1) "For simplicity and to create a challenging transferability test, we train our ML2-based NN only on eight small closed-shell atoms/ions (H-, He, Be, Ne, Mg, Ar, Ca, and Kr)." I iterate that in the comparisons the GES is hopeless since for the GES model the situation is in very overparametrized condition because of too few training data (Authors used the same model architecture with (# of features x 16)+256+256+16 parameters, right?). Meanwhile, the authors' addition Fig. S13 is informative since the improvement of GES is slow if we increase data to alleviate the overparametrization. Note that I'm not doubting that LES is useful but was just saying that the comparison was redundant. But I'm getting inclined to an idea that it may be informative for general readership. I would therefore be satisfied if the fact—in the comparisons GES is suffering from overparametrization (or underdetermined condition, in other words)—is written in a well visible place explicitly. Provided that this point is responded, I will finally recommend Acceptance.

Our answer: The reviewer is right that more explicit statement regarding the GES being prone to overfitting when just these 8 datapoints are used. We address this now towards the end of *Sec. Result/Uniqueness and Robustness of our LES-based ML2 model* in on p. 9 when referencing Fig. S13, following the @2.2 label in the marked-up manuscript.

As a side note, the number of hidden layers in our ML2 is 3 x 16 (the second paragraph of *Methods*) and the results showing the LES robustness when more / less neurons are used are in Fig. 4(d) and S11.

2.3 Point raised by Reviewer: 2) I find ". . . and it what follows" to be fixed.

Our answer: The sentence has been corrected accordingly (see the end of *Sec. Methods/The advantages of LES-based learning process* following the @2.3 label in the marked-up manuscript)

Points raised by Reviewer 3

3.1 Point raised by Reviewer: The authors have addressed all points raised in my previous report and have particularly demonstrated the robustness of their results. I do not have any further questions or comments and recommend publication of their manuscript.

Our answer: We thank the Reviewer for their positive assessment.